# Dynamic Tensor Rematerialization

**Marisa Kirisame,**[*,†] **Steven Lyubomirsky,**[*,†] **Altan Haan,**[*,†] **Jennifer Brennan,**[†]
**Mike He,**[†] **Jared Roesch,**[†,‡] **Tianqi Chen,**[§,‡] **and Zachary Tatlock**[†,‡]
`{jerry96, sslyu, altanh, jrb, dh63, jroesch}@cs.washington.edu,`
`tqchen@cmu.edu, ztatlock@cs.washington.edu`

## Abstract

Checkpointing enables the training of deep learning models under restricted memory budgets by freeing intermediate activations from memory and recomputing them on demand. Current checkpointing techniques statically plan these recomputations offline and assume static computation graphs. We demonstrate that a simple online algorithm can achieve comparable performance by introducing Dynamic Tensor Rematerialization (DTR), a greedy online algorithm for checkpointing that is extensible and general, is parameterized by eviction policy, and supports dynamic models. We prove that DTR can train an $N$-layer linear feedforward network on an $\Omega(\sqrt{N})$ memory budget with only $\mathcal{O}(N)$ tensor operations. DTR closely matches the performance of optimal static checkpointing in simulated experiments. We incorporate a DTR prototype into PyTorch merely by interposing on tensor allocations and operator calls and collecting lightweight metadata on tensors.

## 1 Introduction

As state-of-the-art deep learning (DL) models continue to grow, training them within the constraints of on-device memory becomes increasingly challenging. The memory demands of emerging models prevent their training on memory-limited devices (such as specialized accelerators, low-powered embedded devices, or older GPUs) and limit researchers' ability to explore memory-intensive architectures and training techniques. Checkpointing is one technique that enables training with models and batches that exceed on-device memory without modifying the model's design. It is achieved by freeing some activations from memory and recomputing them on demand. Adapted from techniques in automatic differentiation (Baydin et al., 2015; Griewank & Walther, 2000; Siskind & Pearlmutter, 2018), checkpointing in the DL context exploits the fact that intermediate activations for backpropagation dominate memory usage during training (Sohoni et al., 2019) but can be easily recomputed by replaying parts of the forward pass. Current DL checkpointing techniques (Chen et al., 2016; Jain et al., 2020; Kumar et al., 2019; Gruslys et al., 2016) *statically* plan which activations to recompute offline, requiring an initial stage of model analysis.

In this paper, we demonstrate that static planning is unnecessary for DL checkpointing. We present Dynamic Tensor Rematerialization (DTR), a greedy online algorithm for heuristically checkpointing arbitrary DL models. DTR operates like a tensor-level cache: it collects metadata on tensors and operators as a model is trained and uses it to guide heuristics that choose which activations to free and later recompute. As a runtime system, DTR can utilize dynamically gathered information (*e.g.*, measured operator costs). Additionally, its simple, cache-like approach requires no advance knowledge of the model or application, letting it immediately support arbitrarily dynamic models and applications featuring higher-order differentiation. For example, given a model with data-dependent control flow like TreeLSTM (Tai et al., 2015), DTR's runtime can simply evict tensors when memory runs out and rematerialize them as needed. By contrast, static planning techniques assume a static dataflow graph, which requires "unrolling" dynamic models and performing (potentially expensive) planning for every distinct input.

---

[*]Equal contribution.

[†]Paul G. Allen School of Computer Science & Engineering, University of Washington, Seattle, WA

[‡]OctoML, Seattle, WA

[§]School of Computer Science, Carnegie Mellon University, Pittsburgh, PA

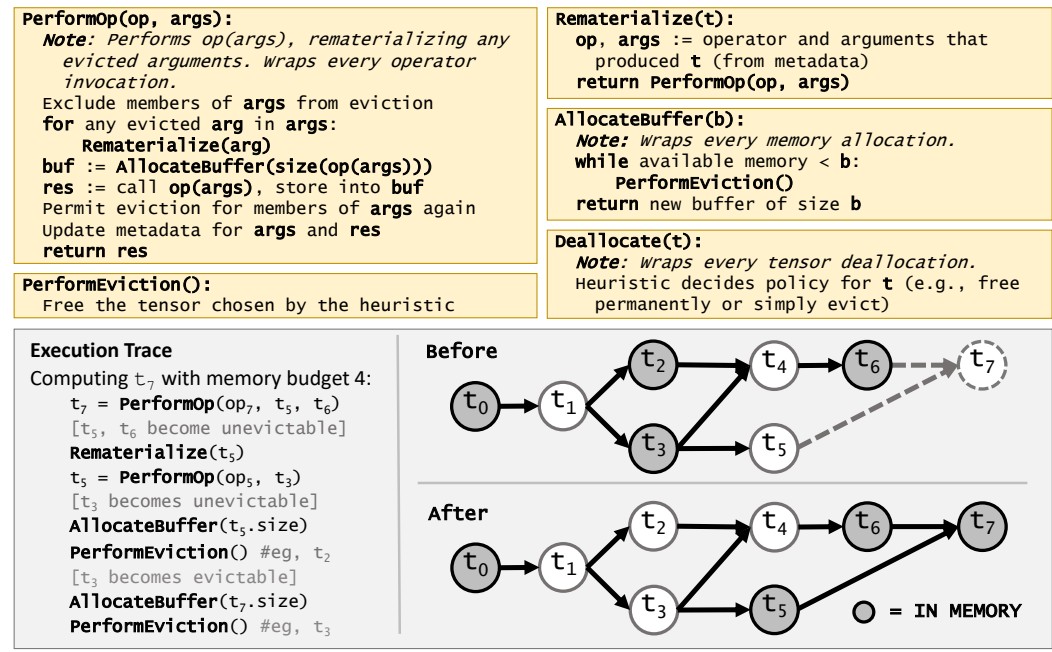

Figure 1: (Top) Pseudocode for DTR's basic logic (independent of heuristic), and (Bottom) DTR's sequence of events in an operator call. Note that `PerformOp()` may make further recursive calls in order to rematerialize arguments.

This paper describes DTR's design (Sec. 2) and makes the following contributions:

- We prove that DTR can train an $N$-layer linear feedforward network on an $\Omega(\sqrt{N})$ memory budget with only $\mathcal{O}(N)$ tensor operations (Sec. 3), which is within a constant factor of optimal and matches the offline bound of the Chen et al. (2016) static checkpointing technique.

- We formalize DL model checkpointing as an online rematerialization problem and define a greedy algorithm parameterized by caching-inspired heuristics. In simulated trials our heuristic attains *near-optimal performance* on a variety of DL models (Sec. 4).

- We implement a DTR prototype by making only modest modifications to the PyTorch framework, enabling training under restricted memory budgets for both static and dynamic models and demonstrating the ease with which our algorithm can be incorporated into an existing DL framework (Sec. 5).

Note that techniques other than checkpointing, such as swapping tensors between devices, can also enable training under limited memory. In Sec. 6, we discuss these approaches and how they could operate with DTR.

## 2 DYNAMIC TENSOR REMATERIALIZATION

We introduce Dynamic Tensor Rematerialization (DTR), a thin runtime layer that intercepts tensor allocations, accesses, and deallocations and eliminates the need for ahead-of-time model analysis to support checkpointing. Figure 1 shows DTR's high-level approach. When a tensor allocation occurs (`AllocateBuffer`), DTR first checks if sufficient memory is available. If so, it generates a fresh tensor identifier, initializes its metadata for future recomputation, allocates the requested memory, and returns a new tensor. If not, DTR heuristically selects and *evicts* resident tensors until the requested allocation can be accommodated. Constant tensors (loaded from external data) cannot be evicted since no corresponding operation rematerializes them. Upon tensor access, DTR first checks if the tensor is resident in memory. If so, it updates tensor metadata before returning the requested tensor. If the tensor has been evicted, DTR *rematerializes* it by replaying the *parent operation* that originally produced the tensor. Crucially, rematerialization can be recursive: if the arguments to an evicted tensor's parent operation have also been evicted, then *they* must first be

rematerialized. Rematerialization may trigger more evictions if memory is exhausted during the potentially recursive process. Upon tensor deallocation (other than by evictions), the runtime is invoked again (`Deallocate`), letting it update tensor metadata and eagerly perform profitable evictions.

**Assumptions.** This description of DTR assumes that: tensors are accessed only by opaque operators; tensors are either constants or produced by operators; operators produce individual tensors; and operators are pure (deterministic functions of their arguments). Under this model, a training epoch is simply a sequence of tensor operations without any inherent requirement to recognize training-specific structure, like the transition to the backward pass. DTR will evict as many tensors as necessary to avoid running out of memory. If all inputs and outputs of a single operation cannot fit into available memory, rematerialization will fail; therefore, on a given model and input, there may be a threshold for the lowest budget DTR can support. The choice of heuristic can affect the likelihood of failure since different eviction choices can result in deeply nested rematerializations that require many tensors to remain in memory.

**Heuristics.** DTR is parameterized by heuristics that guide its eviction choices. As in caching, DTR's eviction heuristic *dynamically* predicts which resident tensors are least valuable. The choice of heuristic determines what metadata (additional runtime facts) must be tracked for each tensor and operator and thus affects DTR's runtime overhead. In our evaluation, we consider a runtime system that tracks the following metadata for each tensor $t$: **staleness**, $s(t)$, the time since last access; **memory**, $m(t)$, the size of the tensor; and **cost**, $c_0(t)$, the time required to compute $t$ from its parent tensor(s). We observe that DTR's metadata overhead is low relative to the cost of typical DL tensor operations.

We propose a rematerialization-specific heuristic that balances staleness, memory, and cost, evicting the tensor $t$ that is stalest (least likely to be needed soon), largest (saves the most space), and cheapest (requires the least additional rematerialization if $t$ is needed again). To capture the total amount of rematerialization required if $t$ is evicted, we sum the costs over the tensor's *evicted neighborhood* $e^*(t)$, *i.e.*, the set of evicted tensors that would either need to be rematerialized to recompute $t$ or would need $t$ to be resident to be recomputed. We define the *projected cost*, $c(t)$, of rematerializing tensor $t$ as $c_0(t) + \sum_{t' \in e^*(t)} c_0(t')$. Using this definition, we define our heuristic, which evicts the tensor minimizing $h_{\text{DTR}}(t) = c(t)/[m(t) \cdot s(t)]$. By including both forward and backward dependencies of $t$ in $e^*(t)$, $h_{\text{DTR}}$ penalizes creating long chains of evicted tensors (and hence potential recursive rematerializations) that could arise from $t$'s eviction.

To illustrate evicted neighborhoods, suppose DTR is checkpointing the network shown in Figure 1, where the resident tensors are $\{t_0, t_2, t_3, t_6\}$. Before node $t_7$ is computed, we have $e^*(t_2) = \{t_1, t_4\}$ and $e^*(t_3) = \{t_1, t_4, t_5\}$. Since each new eviction can expand a given tensor's evicted neighborhood and each rematerialization can shrink it, dynamically tracking evicted neighborhoods can introduce further costs at run time. To decrease runtime overhead, we developed an approximation of $e^*$ using an undirected relaxation tracked by a union-find data structure that uses a constant-time approximation for splitting. We use this approximation to define $h_{\text{DTR}}^{\text{eq}}$ analogously (Sec. 4.1 and Appendix C.3 contain details), which performs nearly as well as $h_{\text{DTR}}$ in our evaluation (Sec. 4) but requires up to 2 orders of magnitude fewer metadata accesses per batch (Appendix D.3).

We compare $h_{\text{DTR}}$ to other heuristics inspired by recent work in our simulated evaluation (Sec. 4) and discuss an even broader class of heuristics in Appendix D. Our heuristic formalization in terms of $s$, $m$, and $c_0$ is sufficiently general to express several existing heuristics for caching and checkpointing. For example, the common LRU heuristic is "minimize $1/s(t)$," the `GreedyRemat` heuristic from Kumar et al. (2019) is "minimize $1/m(t)$," and the MSPS heuristic from Peng et al. (2020) is "minimize $c_R(t)/m(t)$" (where $c_R(t)$ sums $c_0$ over $t$'s evicted ancestors).

**Deallocation.** Deallocation policies present further tradeoffs since tensors marked as deallocated by the original program are still potential dependencies for rematerializations. In principle, DTR could simply disregard deallocations by the original program, but this would ignore potentially useful information about the deallocated tensors (*viz.*, that the original program will not use them again). *Banishing* (permanently freeing) deallocated tensors can save memory immediately and is the only way to free constants (which cannot be evicted); however, it can prevent possible future evictions since the children of a banished tensor cannot be rematerialized. By contrast, evicting deallocated tensors does not prevent potential evictions, though it increases the runtime's management overhead and keeps constants in memory. In the heuristics we examined, we implemented an *eager eviction*

mechanism, which evicts a tensor as soon as all external references to it are freed. This lets DTR adhere to the garbage collection pattern of the underlying framework, preempting desirable evictions, which further reduces future runtime overhead. (See Appendix D.2 for a comparison of deallocation policies.)

## 3 FORMAL BOUNDS

Following Chen et al. (2016), we prove a bound on DTR's checkpointing overhead (for a particular eviction heuristic) on a linear feedforward network of $N$ nodes. Even without the ability to inspect the model, DTR requires only $\mathcal{O}(N)$ tensor operations under a $\sqrt{N}$ memory budget, the same bound (up to constant factors) as the Chen et al. (2016) static checkpointing technique and the optimal $\Theta(N)$ required by a memory-unconstrained algorithm. We also establish that DTR's dynamic approach cannot always match the overhead of static checkpointing: given $N$ tensor operations and a memory budget of $B$, under any deterministic heuristic, an adversary could always construct a network where DTR would perform a factor of $\Omega(N/B)$ more tensor operations than a (potentially expensive, see Jain et al. (2020)) optimal static checkpointing algorithm.

**Linear Feedfoward Overhead.** We assume that tensor computations dominate runtime and, as in prior work (Griewank & Walther, 2000; Chen et al., 2016; Binder et al., 1997; Beaumont et al., 2019b), that each tensor is of unit space and time cost. For the proof below, we use the heuristic $h_{e^*}$, which evicts a resident tensor $t$ with minimal $|e^*(t)|$.

**Theorem 3.1.** *Given an $N$ node linear feedfoward network and a memory budget $B = \Omega(\sqrt{N})$, DTR with heuristic $h_{e^*}$ can execute one forward and one backward pass in $\mathcal{O}(N)$ operations.*

*Proof Sketch.* During the forward pass, DTR performs exactly $N$ tensor operations: since each node of the linear feedforward network depends only on the previous node, no rematerialization is necessary. Our heuristic $h_{e^*}$, which evicts tensors with the smallest evicted neighborhoods, ensures that the $B$ tensors resident at the conclusion of the forward pass are evenly spaced throughout the network. In turn, these evenly spaced checkpoints ensure that DTR never has to successively rematerialize too many tensors. As the backward pass proceeds and checkpoint tensors are freed, the overhead to compute all gradients between the checkpoints $k$ and $k+1$ shrinks as $\log(k)/k^2$, which sums to a constant. The full proof of Theorem 3.1 is provided in Appendix A.

**Adversarial Overhead.** Using a simple heuristic, DTR can match the performance of static checkpointing on linear feedfoward networks despite lacking advance knowledge of the architecture. However, DTR cannot always match the performance of optimal static checkpointing on an arbitrary network because it cannot access or reorder the network.

**Theorem 3.2.** *For any deterministic heuristic $h$, there exists an $N$-node network on which DTR with budget $B \leq N$ requires $\Omega(N/B)$ times more tensor computations than optimal static checkpointing.*

*Proof Sketch.* Generate an adversarial network $G$ of $B$ linear feedforward networks joined by a common parent tensor. Using $h$, schedule $G$'s operations such that, at each step of DTR, the next operation is taken from the end of an entirely evicted path through $G$, forcing DTR to rematerialize the entire path. DTR can thus be forced to perform at least $\Omega(N^2/B)$ operations. By contrast, an optimal static algorithm can reorder $G$ to compute each feedforward network sequentially, requiring only $N$ computations. The full proof of Theorem 3.2 is provided in Appendix B.

Theorems 3.1 and 3.2 illustrate how DTR's performance, from optimal to poor, depends on interactions between heuristics and models. We next explore DTR design tradeoffs empirically.

## 4 HEURISTIC EVALUATION

We simulated DTR on a variety of models to empirically evaluate its checkpointing performance across different heuristics and compare it to the static checkpointing schemes examined in Jain et al. (2020). DTR enables training under restricted memory budgets and closely matches the performance of an optimal baseline.

### 4.1 HEURISTICS EXAMINED

We examine variants of the evicted neighborhood–based $h_{\text{DTR}}$ heuristic described in Sec. 2 (on which we establish formal bounds) as well as heuristics inspired by past work in caching and checkpointing. All following heuristics are defined as a score function in terms of the metadata $m(t)$, $s(t)$, and $c_0(t)$, where the tensor with the minimum score is evicted.

In addition to $h_{\text{DTR}}$, we consider $h_{\text{DTR}}^{\text{eq}}$, which uses an equivalence class–based approximation $\tilde{e}^*$ for $e^*$, and $h_{\text{DTR}}^{\text{local}}$, which only uses individual tensors' costs instead of costs over evicted neighborhoods. We compare against other variants of $h_{\text{DTR}}$ in Appendix D, but here we focus on these in particular because (1) $h_{\text{DTR}}^{\text{local}}$ lets us assess the importance of tracking evicted neighborhoods at run time, and (2) $h_{\text{DTR}}^{\text{eq}}$ lets us evaluate how well $\tilde{e}^*$ approximates $e^*$ in practice. We define the $h_{\text{DTR}}$ variants as:

$$h_{\text{DTR}} \stackrel{\text{def}}{=} \frac{c_0(t) + \sum_{t' \in e^*(t)} c_0(t')}{m(t) \cdot s(t)}, \quad h_{\text{DTR}}^{\text{eq}} \stackrel{\text{def}}{=} \frac{c_0(t) + \sum_{t' \in \tilde{e}^*(t)} c_0(t')}{m(t) \cdot s(t)}, \quad h_{\text{DTR}}^{\text{local}} \stackrel{\text{def}}{=} \frac{c_0(t)}{m(t) \cdot s(t)}.$$

Rather than using directed dependencies, $\tilde{e}^*(t)$ treats the dependency graph of tensors as undirected (thus admitting some spurious dependencies), letting us decompose the graph into a set of disjoint evicted components. We can track these evicted components efficiently using a union-find data structure with a running sum for each component. When a tensor $t$ is evicted, its component is unioned with those of any evicted neighbors and $c_0(t)$ is added to the component's running sum. Though this enables near-constant-time merging between components (by unioning and adding the sums), union-find does not support splitting. To efficiently split components, we make another approximation: when a tensor $t$ is rematerialized, we simply subtract $c_0(t)$ from its component's running sum and map $t$ to a new (empty) union-find component. Since this approach removes no connections, it produces "phantom dependencies" between some tensors. In practice, we find that despite these additional dependences, $h_{\text{DTR}}^{\text{eq}}$ closely matches the performance of $h_{\text{DTR}}$ (Figures 2 and 3) but requires fewer operations per eviction and rematerialization. See Appendix C.3 for a more detailed description of $\tilde{e}^*(t)$.

We also consider the following heuristics inspired by past work:

$$h_{\text{LRU}}(t) \stackrel{\text{def}}{=} \frac{1}{s(t)}, \quad h_{\text{size}}(t) \stackrel{\text{def}}{=} \frac{1}{m(t)}, \quad h_{\text{MSPS}}(t) \stackrel{\text{def}}{=} \frac{c_0(t) + \sum_{t' \in e_R(t)} c_0(t')}{m(t)},$$

where $e_R(t)$ is the set of evicted tensors that would have to be rematerialized in order to rematerialize $t$. $h_{\text{LRU}}$ is based on the common "least-recently used" policy for caching, $h_{\text{size}}$ is based on `GreedyRemat` from Kumar et al. (2019) (used in TensorFlow XLA), and $h_{\text{MSPS}}$ is based on the MSPS heuristic from Peng et al. (2020). We also include a random baseline, $h_{\text{rand}}(t) \stackrel{\text{def}}{=} X \sim U(0,1)$, to assess how well a heuristic using no metadata whatsoever performs.

### 4.2 COMPARING DTR ACROSS HEURISTICS

**Experimental Setup.** To model a realistic execution setting for DTR, we instrumented Py-Torch (Paszke et al., 2019) to log operations performed, metadata on tensors and operators (including sizes, compute times, and parent tensors), and deallocations during the execution of various models. We replayed the logs in a simulator that models the behavior of DTR in the style shown in Figure 1. The simulator tracks the tensors in memory at any given time, chooses tensors to evict per the heuristic when the memory budget is exceeded, and sums the total cost of the model operators and rematerializations. For verisimilitude, the simulator also models the semantics of various low-level PyTorch implementation details, including tensor aliasing, in-place mutation, and multi-output operations. We gathered logs from several static models examined in recent work, such as Jain et al. (2020) and Peng et al. (2020), in addition to three dynamic models (LSTM, TreeLSTM, and Unrolled GAN); each log corresponds to an execution of the forward pass, computing the loss, and performing the backward pass. The simulator also enforces the additional condition that gradients for all trainable weights be resident at the end of the simulation in order to model the requirements for performing a full training step. Appendix C gives a full technical specification of the simulator and log format.

**Results.** For all models in Figure 2, DTR executed a training step using a small fraction of the normal memory required with limited compute overhead. Furthermore, unlike existing static approaches,

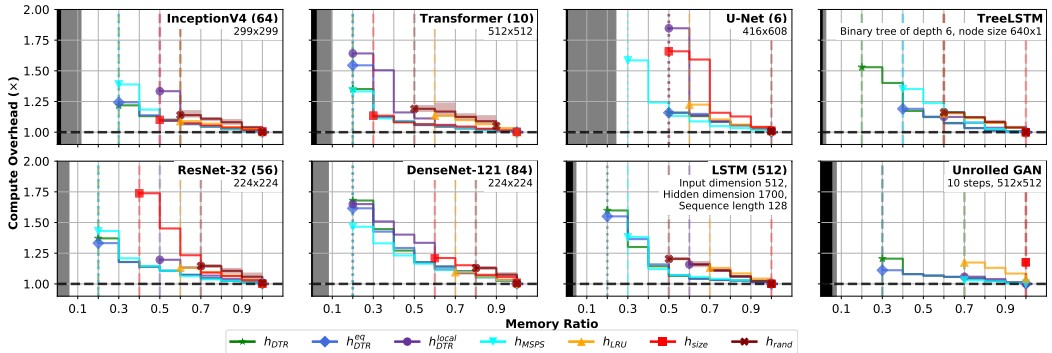

Figure 2: Simulated results comparing different heuristics on various models, showing the rate of computational slowdown for different budgets (fractions of the original peak memory usage). The black area in each graph corresponds to the memory required to store inputs and weights, while the gray area denotes the single operator requiring the most memory to be live at once. The dashed and dotted lines represent the last ratio before thrashing ($\geq 2\times$ slowdown) and out-of-memory errors, respectively. All logs were produced by running each model 50 times on a single input on a machine with an NVIDIA Titan V GPU (CUDA 10.1, CuDNN 7.6.4) and a 16-core AMD Ryzen Threadripper 1950X on Ubuntu 18.04, logging the final "warmed-up" run.

DTR automatically supports models with arbitrary dynamism. In all cases, results show that heuristics incorporating more information about chain rematerializations ($h_{\text{DTR}}$, $h_{\text{DTR}}^{\text{eq}}$, and $h_{\text{MSPS}}$) can operate on lower budgets and perform fewer rematerializations than heuristics using less information. However, these complex heuristics also introduce more *runtime* overhead, which must be considered when implementing DTR. In particular, our simulations showed that $h_{\text{DTR}}$ incurred up to 2 orders of magnitude more metadata accesses per batch compared to $h_{\text{DTR}}^{\text{eq}}$, and up to 3 orders of magnitude more compared to $h_{\text{DTR}}^{\text{local}}$ (see Appendix D.3). The fact that $h_{\text{DTR}}^{\text{eq}}$ closely matches the performance of $h_{\text{DTR}}$ while incurring much less runtime overhead suggests that it would be more effective in practice. Note that even simple heuristics like $h_{\text{LRU}}$, which require only modest runtime overhead, typically enabled training with 30% less memory.

## 4.3 COMPARING DTR TO STATIC TECHNIQUES

We compared the performance of DTR using $h_{\text{DTR}}$, $h_{\text{DTR}}^{\text{eq}}$, and (as a simple baseline) $h_{\text{LRU}}$ against static checkpointing techniques, including the optimal Checkmate tool of Jain et al. (2020). As Figure 3 shows, DTR's $h_{\text{DTR}}$ and $h_{\text{DTR}}^{\text{eq}}$ heuristics obtain performance remarkably close to Checkmate's optimal solutions; even the much simpler $h_{\text{LRU}}$ heuristic obtains superior performance relative to the static baselines. While Checkmate requires full ahead-of-time knowledge of the model and seconds or minutes per budget to compute guaranteed-optimal solutions using an integer linear programming (ILP) solver, *DTR finds comparable solutions dynamically and in milliseconds* without ahead-of-time knowledge of the model.

## 5 PROTOTYPE IMPLEMENTATION

We implemented a DTR prototype[1] in PyTorch and evaluated its performance on a variety of models. We chose PyTorch because its eager mode of execution ("define by run") accomodates arbitrary control flow in models but makes static analysis more difficult; hence, it is a setting where DTR's online nature is an asset. Per the results in Sec. 4, we implemented $h_{\text{DTR}}^{\text{eq}}$ as the prototype's heuristic. *The core system was implemented in only 1,161 lines of code and made no deep modifications to PyTorch's memory management internals or tensor abstractions, illustrating the simplicity of our system.* The remaining 2,647 lines of changes were primarily boilerplate operator overloads used to dispatch tensor operations through DTR's core logic (Appendix E.1 describes our prototype implementation's structure).

---

[1]Publicly available at `https://github.com/uwsampl/dtr-prototype`

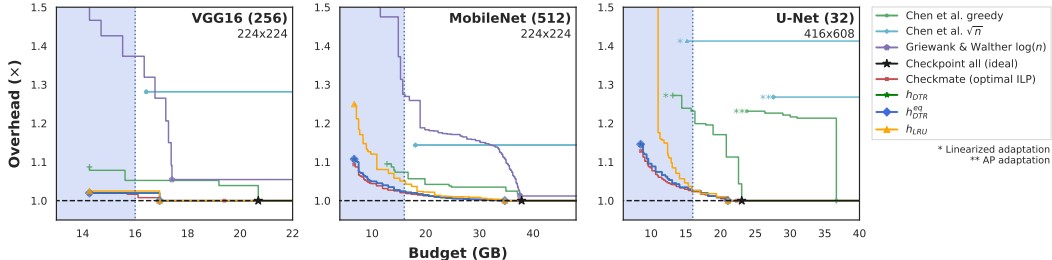

Figure 3: DTR's overhead from operators is competitive with Checkmate's, which uses ILP to produce an optimal rematerialization schedule. This comparison extends Figure 5 in Jain et al. (2020) by adding the DTR simulator as a "solver" that translates Checkmate's Keras-based graph representation into the DTR simulator's representation. To produce this comparison, we modified Jain et al. (2020)'s evaluation artifact because the PyTorch logs from Sec. 4.1 did not contain some information that past checkpointing techniques require (such as which backward operators correspond to which forward ones). Also included in the comparison (from the original experiment) are the Griewank & Walther (2000) Treeverse algorithm and variants of the Chen et al. (2016) checkpointing algorithm (modified to handle skip connections like those in ResNet).

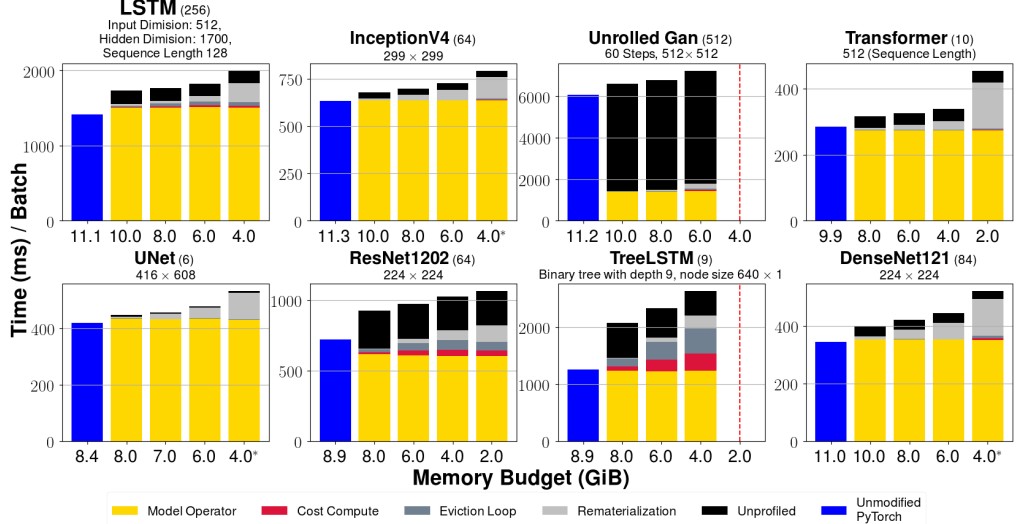

Figure 4: We profiled the running time of our prototype for various models and memory budgets on a machine with an NVIDIA Titan V GPU (CUDA 10.1, CuDNN 7.6.4) and a 16-core AMD Ryzen Threadripper 1950X on Ubuntu 18.04. The red dotted lines correspond to trials that either ran out of memory or thrashed ($\geq 2\times$ unmodified PyTorch's time). Model batch sizes are given in parentheses. To ensure the accuracy of the DTR prototype's profiling, we used PyTorch's synchronous computation mode (see Appendix E.1). Results (mean of 100 trials) are compared against unmodified PyTorch. "Cost compute" (computing heuristic scores) and "eviction loop" (comparing scores over tensors) correspond to overhead from the DTR *runtime* itself, which can be reduced by a more efficient implementation. "Unprofiled time" is the remainder of the time per batch; it may be due to runtime overhead from parts of PyTorch not modified in the prototype, like the operator dispatch system. The large proportion of unprofiled time in Unrolled GAN is likely due to its extensive use of Python reflection. The budgets with asterisks were run with the random sampling optimization (see Appendix E.2) disabled, as sampling caused occasional failures at those budgets.

Our empirical evaluation demonstrates that DTR can efficiently train models under restricted memory budgets using the $h_{\text{DTR}}^{\text{eq}}$ heuristic. We used the same models and experimental setup as in Section 4, timing the forward pass, loss computation, and backward pass. Table 1 presents several cases where DTR trains models on much larger input sizes than unmodified PyTorch, including a dynamic model,

| | ResNet-1202 (Batch Size) | | | | Transformer (Batch Size) | | | | UNet (Batch Size) | | | | TreeLSTM (Tree Nodes) | | | |
|---|---|---|---|---|---|---|---|---|---|---|---|---|---|---|---|---|
| | 64 | 100 | 120 | 140 | 30 | 70 | 80 | 90 | 7 | 8 | 9 | 10 | $2^6$-1 | $2^7$-1 | $2^8$-1 | $2^9$-1 |
| DTR | 0.974s | 1.18s | 1.28s | 1.39s | 367ms | 830ms | 950ms | 1079ms* | 566ms | 684ms | 822ms* | 1170ms* | 0.486s | 1.05s | 2.50s | 7.89s* |
| PT | 0.712s | ✗ | ✗ | ✗ | 331ms | ✗ | ✗ | ✗ | 481ms | ✗ | ✗ | ✗ | 0.431s | ✗ | ✗ | ✗ |

Table 1: Median execution times per batch (out of 100 runs) for various models, giving both the largest input size that unmodified PyTorch ("PT") could support on our GPU and larger input sizes DTR could support. Input sizes are as in Figure 4, except for TreeLSTM (complete binary trees with nodes of size $1024 \times 1024$) and Transformer (sequence length 256). Asterisks indicate inputs on which the random sampling optimization was disabled due to occasional failed trials. Even without sampling, DTR still occasionally failed on UNet (see Appendix E.3 for details). This behavior may be due to PyTorch memory allocator implementation details or poor rematerialization decisions influenced by variance in individual operator times.

TreeLSTM. This highlights that *DTR enables exploration of models that push the boundaries of existing deep learning architectures*. While the simulated trials in Sec. 4.2 consider the slowdown due only to rematerializations but not overhead from managing metadata and computing heuristics, Figure 4 measures the time per batch required to train eight DL models on a variety of restricted memory budgets, profiling the time spent by the runtime system. Among the models is Unrolled GAN, which uses higher-order partial derivatives and Python reflection extensively; *the DTR prototype supported these unusual features, underscoring its generality*. Despite our prototype's simplicity — it merely loops through all tensors when searching for an eviction candidate and recomputes the heuristic scores from scratch each time — on most models, *its overhead due to searching and computing heuristics remains low for most memory budgets*. In Appendix E.2, we discuss two approximate optimizations we included in the prototype to reduce the overhead of searching over tensors and additional ways to reduce DTR's runtime overhead.

# 6 RELATED WORK

**Checkpointing in Reverse-mode Automatic Differentation (AD).** Checkpointing in DL takes inspiration from checkpointing in reverse-mode AD (Baydin et al., 2015). The latter reduce the number of values stored in the "tape" by recomputing segments of the tape (demarcated by "checkpoints"). Treeverse (Griewank, 1994; Griewank & Walther, 1997; 2000) uses a binomial partitioning scheme to mark checkpoints, achieving logarithmic growth in space in exchange for a logarithmic grown in computation. Later works, such as Hascoet & Pascual (2013) and Siskind & Pearlmutter (2018), extend Treeverse's approach to handle arbitrary control flow by inserting code at compile time to mark checkpoints according to policies (*e.g.*, "checkpoint every $k$ iterations" for a statically unbounded loop). Unlike DTR, these techniques do not use dynamically gathered information.

**Checkpointing in DL.** Many DL models can be represented as static dataflow graphs, enabling the straightforward application of Treeverse-like partitioning approaches. Chen et al. (2016) apply this approach by dividing the network into segments to be recomputed during backpropagation, presenting schemes that allow for training an $N$-layer feedforward network in $\mathcal{O}(\sqrt{N})$ memory with one extra forward pass ($\mathcal{O}(N)$ tensor operations) or in $\mathcal{O}(\log N)$ memory with $\mathcal{O}(N \log N)$ additional tensor operations. Gruslys et al. (2016) present a similar segmenting approach for recurrent neural networks, thereby supporting some dynamism beyond static computation graphs. Other recent work rematerializes individual activations rather than entire segments, attaining better bounds than Chen et al. (2016); Kusumoto et al. (2019), Kumar et al. (2019), and Beaumont et al. (2019a) apply graph-theoretic analyses to make rematerialization plans, while Jain et al. (2020) apply integer linear programming (ILP) to find optimal solutions.

DTR differs fundamentally from those approaches because it handles arbitrary dynamic control flow in models (making no assumptions about the model's structure) and operates online, giving it access to dynamically gathered information. In principle, a static checkpointing technique could be applied to a dynamic model "just in time" by unrolling the model on the fly, but some static analyses (like an ILP solver) can be too expensive to run each epoch. Unlike static approaches, however, dynamic planning introduces overhead at run time, which limits the analyses that DTR's heuristics can feasibly perform. Note that the Chen et al. (2016) greedy scheme and the `GreedyRemat` baseline in Kumar et al. (2019) are similar to DTR in that they greedily place checkpoints using a heuristic (albeit statically). However, their heuristics only use the sizes of tensors.

**DL Memory Managers.** Other work has enable the training of DL models on lower memory budgets by swapping tensors between GPUs or to host RAM. Huang et al. (2020) use a genetic algorithm to plan swaps between devices on static computation graphs. Capuchin by Peng et al. (2020) and Superneurons by Wang et al. (2018), like DTR, use runtime systems and incorporate checkpointing as well. Capuchin's checkpointing phase, which resembles DTR's, uses dynamically gathered information for checkpointing; it performs a single batch without checkpointing (only swapping) and uses the costs it measures to determine where to set checkpoints. However, Capuchin's and Superneurons's checkpointing schemes assume a static model architecture (inferred from an initial profiling batch), which they use to plan recomputations in advance. Swapping systems like Capuchin rely on interleaving communication and computation at a low level for performance, which may be difficult to apply in an online setting.

These works highlight that swapping and rematerialization are complementary approaches, raising the question of whether DTR can be combined with swapping without disrupting existing methods' overlapping of computation and communication. One possibility would be to assume a fixed swapping schedule and use DTR to replace the rematerialization schemes used by systems like Capuchin (perhaps given a constraint like treating values to be swapped out as unevictable). Another intriguing possibility would be to use swapping as a form of "eviction" in DTR, where the "cost" for swapped-out values would be the communication time. Swapping presents interesting tradeoffs with rematerializations since it may scale better than some tensor operators. However, incorporating swapping into DTR's online approach presents the problem of efficiently overlapping computation and communication since the runtime would need to guarantee that a computation scheduled concurrently with a swap would not need to swap values back in. This could greatly complicate planning (*e.g.*, requiring some lookahead to avoid missed swapping opportunities) and would be fertile ground for future work.

**Memory-Efficient DL Model Designs.** Some recent work manually modifies DL models to perform similar computations using less memory, which may be used alongside checkpointing and swapping approaches. One example is the use of reversible layers, which enable recomputing a forward value during backpropagation using the result of the following layer. Gomez et al. (2017) and Kitaev et al. (2020) employ reversible layers to create versions of ResNet and Transformer, respectively, that can train using less memory.

## 7 CONCLUSION

DTR provides a simple, customizable approach to checkpointing for DL models. It supports a broad range of applications without the need for any ahead-of-time analyses, manual annotations, or modifications. Our formal results establish that DTR can match the same asymptotic bounds as recent static checkpointing approaches for linear feedforward networks. In simulation, it enables training for a range of both static and dynamic models under various restricted memory budgets and closely matches the performance of optimal checkpointing. The DTR prototype in PyTorch demonstrates how our approach can be incorporated into existing frameworks with modest, non-invasive changes by simply interposing on tensor allocations and operator calls and collecting lightweight metadata on tensors. Our results also open several avenues for future work. For example, DTR could easily be extended to leverage additional information that may further reduce runtime overhead, such as learning from past batches.

## ACKNOWLEDGEMENTS

This work was supported by the Applications Driving Architectures (ADA) Research Center, a JUMP Center co-sponsored by SRC and DARPA. The Titan V used for this research was donated by the NVIDIA Corporation. We thank Paras Jain and Aniruddha Nrusimha for assistance in setting up and running the Checkmate MLSys 2020 artifact and providing helpful additional information about the Checkmate tool. We are grateful to Edward Z. Yang for helpful advice on modifying PyTorch. We acknowledge Yonghao Zhuang for drawing our attention to an omission in our description of the $h_{\mathrm{DTR}}^{\mathrm{eq}}$ splitting approximation in Section 4.1, which we have corrected in this version. We also thank Sandy Kaplan, Eunice Jun, Josh M. Pollock, Samuel Ainsworth, and Sam Kaufman for providing feedback and useful comments on various drafts of this work.

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

## A  PROOF OF THEOREM 3.1

In this section, we provide a proof of the $\mathcal{O}(N)$ runtime of DTR on a linear feed-forward network with uniform operator compute and memory cost, under a reduced heuristic. We begin with a thorough treatment of the network architecture, and then motivate our reduced heuristic $h_{e^*}$ in this simplified setting. Finally, we prove Theorem 3.1.

### A.1  NETWORK DEFINITION

We assume the network consists of operators $f_1, \ldots, f_N$, where the tensor computed by the $i$th operator is given by $f_i(t_{i-1})$, with $t_j$ denoting the tensor computed by the $j$th operator. Note that we consider $t_0$ to be the input tensor, which for simplicity will always reside in memory and not contribute to the active memory consumption. For this reason, we may consider $f_1$ to be a nullary operator. Additionally, we assume that the size of each tensor (denoted $m(t)$) is 1, and likewise for the compute time $c_0(f_i)$ for each operator $f_i$. Note that we may write $c_0(f_t)$ to mean the same as $c_0(f_i)$ for $t = t_i$, when the index $i$ is not convenient.

For backpropagation, we assume each operator $f_i$ has an associated *gradient* operator $\hat{f}_i$, which computes the result $\hat{t}_i = \hat{f}_i(t_{i-1}, \hat{t}_{i+1})$. We may consider $\hat{t}_{N+1} = \mathbf{1}$ to be an unevictable unit tensor, as is the case in automatic differentiation, but for simplicity we define $\hat{t}_1 = \hat{f}_1(\hat{t}_2)$ and $\hat{t}_N = \hat{f}_N(t_{N-1})$. As above, we assume unit memory and compute for each $\hat{f}_i$.

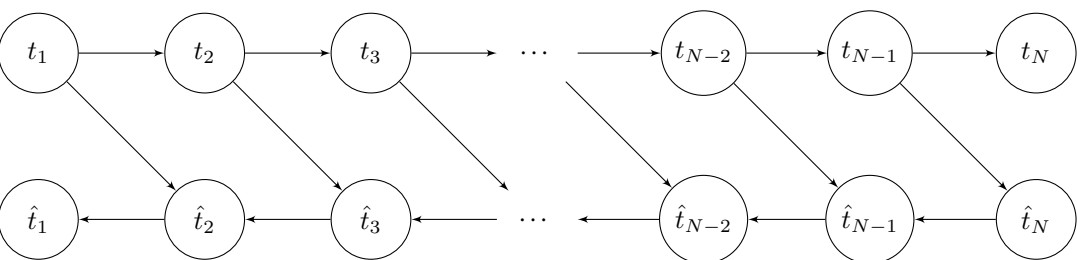

### A.2  LIVENESS AND BANISHING

To optimize memory usage during computation, we introduce the notion of *liveness* and *banishing*. At a high level, liveness allows us to determine when a given tensor is no longer required for subsequent network computations, which in turn allows us to permanently free (banish) tensors to regain memory when certain conditions are met.

To be more precise, we formalize the network as a program:

```
let t₁  := f₁();
let t₂  := f₂(t₁);
...
let t_N  := f_N(t_{N-1});
// Backpropagate.
let t̂_N  := f̂_N(t_{N-1});
let t̂_{N-1}  := f̂_{N-1}(t_{N-2}, t̂_N);
...
let t̂₂  := f̂₂(t₁, t̂₃);
let t̂₁  := f̂₁(t̂₂);
```

We say a tensor $t$ is *live* when there is a pending operation in the program that takes $t$ as an input. When $t$ is no longer live, *and* every tensor directly computed using $t$ is in memory or banished, then we say $t$ is *banished* and we reclaim the memory used by $t$. Banishing a tensor additionally makes its children unevictable.

Thus for example, $t_N$ can be immediately banished after computing, $t_{N-1}$ can be banished after $\hat{t}_N$, both $t_{N-2}$ and $\hat{t}_N$ after $\hat{t}_{N-1}$, and so on. This will become important in the proof.

The analysis of liveness can be done statically for static models, and by reference counting for models with dynamism. In both cases, liveness information is fed to DTR online through deallocation events.

## A.3   HEURISTIC DEFINITION

Heuristic $h_{e^*}$ is a reduced form of the DTR heuristic, as it does not account for tensor staleness. Here, we provide a detailed motivation of its definition.

Recall the *evicted neighborhood* $e^*(t)$ of tensor $t$, as described in Section 2 and further formalized in Appendix C.2.

**Definition A.1** (Projected Cost). For a given tensor $t$, the *projected cost* of $t$ is the value

$$c(t) = \sum_{t' \in e^*(t)} c_0(f_{t'})$$

Now, we define the reduced heuristic in full generality; the definition of $h_{e^*}$ will be a consequence of the simplified setting we analyze.

**Definition A.2** (Compute-Memory Heuristic (general)). The *compute-memory* heuristic score for a resident tensor $t$ is defined as

$$h_{e^*}(t) = \frac{c(t) + c_0(f_t)}{m(t)}$$

**Corollary A.1.** *Under our simplified compute and memory constraints, $h_{e^*}(t) = |e^*(t)| + 1$. Since the heuristic is only used to rank tensors, the common additive constant $1$ is unimportant. The heuristic $|e^*(t)|$ will have the same behavior as $|e^*(t)| + 1$.*

Note importantly that uncomputed tensors are not considered in any of the above definitions (as we do not know about their existence yet, from a dynamic execution perspective).

## A.4   PROOF OF THEOREM 3.1

Now we prove Theorem 3.1, which bounds the overhead of DTR on a linear feedforward network with $N$ nodes and $\sqrt{N}$ memory by a constant factor of the runtime required by an algorithm with unlimited memory.

*Proof.* To prove this claim, we will consider the forward pass and the backward pass separately. In the forward pass, we show that our algorithm only performs $N$ computations, matching that of an algorithm with unlimited memory. Furthermore, upon completion of the forward pass, we tightly characterize the $B$ tensors that remain in memory. We show that a set of evenly spaced *checkpoint tensors* remain in memory throughout the backward pass, until banishment. The presence of these checkpoint tensors allows us to argue that the algorithm never has to rematerialize too many tensors in a row. Furthermore, as the algorithm computes additional gradients, it banishes checkpoint tensors that are no longer needed, freeing more space for additional checkpoints. The overhead incurred by the algorithm can therefore be kept to a constant factor of the required $\Theta(N)$ time. This checkpointing behavior can be seen in the trace of the algorithm, visualized in Figure 5.

We now analyze each of the phases in detail.

PHASE 1: FORWARD PASS

Recall that in a feed-forward network, every computation depends only on the preceding one. Thus in our simplified network, we only ever need $B = 2$ units of memory to compute the forward pass without any rematerializations (furthermore, this is the minimum required memory). For this reason, the forward pass requires $N$ computations.

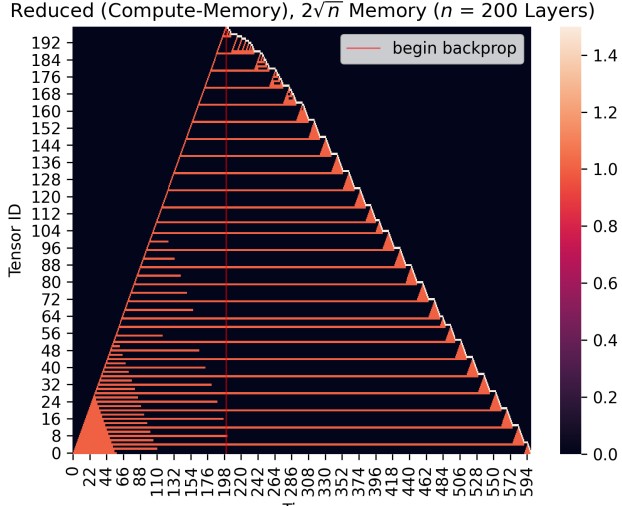

Figure 5: Visualization of the state of memory for DTR with $N = 200$, $B = 2\lceil\sqrt{N}\rceil$, and heuristic $h_{e^*}$. A value of 0 (black) indicates the tensor is evicted or banished, 1 (red) indicates the tensor is a forward value in memory, and 1.5 (white) denotes an in-memory gradient tensor corresponding to the forward tensor. The backward pass begins at the red vertical line; note the presence of evenly spaced *checkpoint tensors* (red horizontal lines) that persist in memory throughout the backward pass. Note also the recursive checkpointing behavior visible in the early gaps of the backward pass, and finally the completely red triangles of the later gaps, when there is enough free memory to avoid repeated rematerialization altogether.

After completing the forward pass, we can tightly characterize the tensors remaining in memory. In particular, Lemma A.1 tells us that the maximum gap between resident tensors is bounded by

$$L \leq \frac{2(N-2)}{B-1}$$

We note that this bound is tight in an asymptotic sense: if we can keep $B$ tensors in memory, and the forward pass is of length $N$, then the maximum gap must be at least $N/B$.

Next, we will analyze the backward pass. Key to this analysis is the claim that "not too many" of the tensors in memory at the beginning of the forward pass are evicted before banishment during the backward pass. The existence of these "checkpoint tensors" allows us to argue that we do not do too much rematerialization work.

PHASE 2: BACKWARD PASS

During the backward pass, our algorithm computes gradients $\hat{t}_i$. Each gradient computation relies on two inputs: $\hat{t}_{i+1}$ and $t_{i-1}$. We show that neither input incurs too much rematerialization cost - $\hat{t}_{i+1}$ because it is pinned in memory, and $t_{i-1}$ because the paths of evicted tensors are not "too long." The first condition follows from the fact that $t_i$ is banished after computing $\hat{t}_{i+1}$, therefore forcing $\hat{t}_{i+1}$ to remain in memory until it is banished. The second condition is formalized in the following lemma, proved later in this section.

**Lemma A.1** (Checkpointing). *Consider an execution of the DTR algorithm with $B$ units of memory and heuristic $h_{e^*}$, applied to the graph described in section A.1. Let $S$ be the set of tensors in memory after computation of $t_N$ in the forward pass. Then, $C \subseteq S$ is a set of "checkpoint" tensors from the forward pass with the following properties:*

   *1. During the backward pass, each $c \in C$ stays in memory until it is banished.*

*2. The gap between neighboring tensors in $\mathcal{C}$ satisfies*

$$L \leq \frac{4(N-2)}{B-1}$$

These $|\mathcal{C}|$ checkpoint tensors divide the $n$ forward tensors into $|\mathcal{C}|$ groups, indexed by $k$, each of length $L_k \leq \frac{4(N-2)}{B-1}$. The total computational cost of the backward pass is equal to the sum of the computational cost for each group,

$$C = \sum_{k=1}^{|\mathcal{C}|} C_k.$$

The second key insight in the analysis of the backward pass is that, for every group that is processed, the algorithm banishes a checkpoint tensor $c \in \mathcal{C}$ and receives a unit of extra memory. In particular, at the start of processing group $|\mathcal{C}| - k$, the algorithm has $2 + k$ pieces of extra memory (two from banishing the most recently used gradient and forward tensor, and $k$ from the banished checkpoint tensors). We can leverage this extra memory to process the gradients in later groups with less rematerialization overhead, using the $k$ extra units of memory to create intermediate checkpoint tensors. The following lemma describes how the cost of computing all the gradients in a group decreases as we free more memory.

**Lemma A.2.** *Suppose we have $2+k$ pieces of free memory to compute all of the gradients associated with an evicted forward tensor path of length $L_k$. Then the number of rematerializations needed to compute all the gradients is of order*

$$C_k = \mathcal{O}\left(L_k + \frac{L_k^2}{k^2}\log k\right)$$

Applying this lemma, the total cost of the backward pass becomes

$$C = \sum_{k=1}^{|\mathcal{C}|} C_k$$

$$\lesssim \sum_{k=1}^{|\mathcal{C}|} \left(L_k + \frac{L_k^2}{k^2}\log k\right)$$

$$\leq \sum_{k=1}^{|\mathcal{C}|} L_k + \sum_{k=1}^{|\mathcal{C}|} \frac{\log k}{k^2} L_k^2$$

$$\leq |\mathcal{C}|\left(\frac{4(N-2)}{B-1}+1\right) + \sum_{k=1}^{|\mathcal{C}|} \frac{\log k}{k^2}\left(\frac{4(N-2)}{B-1}+1\right)^2$$

$$\lesssim |\mathcal{C}|\left(\frac{N}{B}\right) + \frac{N^2}{B^2}\sum_{k=1}^{|\mathcal{C}|} \frac{\log k}{k^2}$$

where $\lesssim$ hides constant factors. Note that $|\mathcal{C}| \leq B$, since $\mathcal{C} \in S$ where $S$ is the set of tensors in memory at the end of the forward pass. Also note that $\frac{\log k}{k^2}$ is a convergent sequence, so its partial sums are bounded. Therefore, we can simplify the bound to

$$C \lesssim N + \frac{N^2}{B^2}$$

Since $B = \Omega(\sqrt{N})$, we conclude that the total cost of the backward pass is $\mathcal{O}(N)$. Adding this to the $\mathcal{O}(N)$ cost of the forward pass, we see the total compute is $\mathcal{O}(N)$, as desired.

$\square$

### A.5 PROOFS OF INTERMEDIATE RESULTS

Here, we present intermediate results that we used in the proof of our main result.

**Lemma A.3.** *Consider the DTR algorithm operating with heuristic $h_{e^*}$. Suppose we seek to (re)materialize forward tensor $t_k$ for $k \leq N$, where the resident tensor preceding $t_k$ is denoted by $t_j$ (with $j < k$). Suppose also that $t_j$ is not evicted during the computation of $t_k$. Then, if the algorithm begins with $t_j$ in memory and with $M$ units of memory, and runs until computing $t_k$, then the maximum length $L$ of any evicted sequence of tensors between $t_j$ and $t_k$ is bounded by*

$$L \leq 2((k-j) - 1)/(M-1)$$

*Proof.* Proof by induction. We will show that, when the algorithm computes tensor $j + i$, for $i = 1, 2, \ldots, k - j$, the maximum length of an evicted sequence of tensors between $t_j$ and $t_{j+i}$ satisfies

$$L_i \leq 2(i-1)/(M-1)$$

**Base case**. When $i = 1$, both $t_j$ and $t_{j+1} = t_k$ are resident tensors, so the gap is $L_1 = 0$.

**Inductive step**. Consider the contents of memory after computing $t_{j+i}$. We begin by partitioning tensors $t_j, \ldots, t_{j+i}$ into $M$ segments $S_1, \ldots, S_M$, each ending in a resident tensor (note, the last segment must end on a resident tensor, since $t_{j+i}$ was just computed). If $i < M$ so that there are not $M$ resident tensors, then the length of each segment is zero and we are done. Otherwise, each segment corresponds to an evicted sequence of zero or more tensors (i.e., the tensors preceding the resident tensor). Let $s_i$ denote the resident tensor that ends segment $i$.

Now, consider all adjacent pairs of segments $(S_l, S_{l+1})$ for $1 \leq l \leq M - 1$. The average length of the pairs is given by

$$\overline{L} = \sum_{l=1}^{M-1} \frac{|S_l| + |S_{l+1}|}{M-1}$$

$$= \left( 2 \sum_{l=1}^{M} \frac{|S_l|}{M-1} \right) - \frac{|S_1| + |S_M|}{M-1}$$

$$= \frac{2}{M-1} \left( \sum_{l=1}^{M} |S_l| \right) - \frac{|S_1| + |S_M|}{M-1}$$

$$= \frac{2i}{M-1} - \frac{|S_1| + |S_M|}{M-1}$$

$$\leq \frac{2(i-1)}{M-1}.$$

Let $(S_{l'}, S_{l'+1})$ be the pair of adjacent segments with minimum combined length. Since the average length is bounded by the inequality above, it follows that the length of $(S_{l'}, S_{l'+1})$ is also less than or equal to $2(i-1)/(M-1)$.

Since the heuristic evicts the tensor that results in the smallest gap, we conclude that the eviction will create a gap no larger than $2(i-1)/(M-1)$. By the inductive hypothesis, the largest previous gap was no larger than $2(i-2)/(M-1)$, so we conclude that the largest gap after this computation is no more than $2(i-1)/(M-1)$.

$\square$

### PROOF OF LEMMA A.1

*Proof.* We will prove this lemma by dividing the backward pass into two phases. In the first phase, the first two gradient computations of the backward pass, we may be forced to evict some element of $S$. In the absence of further information on the evicted tensor, we upper bound the resulting gap by twice the maximum gap between tensors in $S$. This gives us the upper bound in Item 2 of the lemma.

In the second phase, the remaining $N - 2$ gradient computations of the backward pass, we show that heuristic $h_{e^*}$ never leads us to evict a tensor that would lead to a gap of more than $\frac{4(N-2)}{B-1}$ among the tensors in memory. This allows us to conclude that the checkpoint tensors $\mathcal{C}$ remain in memory until eviction, as claimed.

We now elaborate on the two phases, as discussed above.

**Phase 1: The first two gradient computations of the backward pass.**

We present a detailed treatment of the first two gradient computations in the backward pass, $\hat{t}_N$ and $\hat{t}_{N-1}$. We will show that, during the course of these two computations, at most one tensor from $S$ is evicted from memory. Since Lemma A.3 tells us that the maximum gap in $S$ satisfies $L_S \leq \frac{2(N-2)}{B-1}$, we conclude that removing a single tensor results in a gap in $C$ of no more than $2L_S$. Additionally, we will show that after the computation of the first two gradients, there are at least two non-checkpoint tensors in memory. Since only two free units of memory are required to rematerialize a path of tensors, this sets us up for the analysis of the remaining gradient computations.

We begin by noting that, after the forward pass completes, $t_N$ and $t_{N-1}$ are both in memory (since $t_N$ has just been computed, which requires $t_{N-1}$). Since $t_N$ is no longer needed in subsequent computations, it is immediately banished. Assuming $B \leq N$, this leaves us with exactly one unit of free memory (if $B > N$, no elements of $S$ are banished in the first two computations, and the $2L_s$ bound is trivial). This single unit of memory is then filled by the computation of $\hat{t}_N$, which only depends on $t_{N-1}$.

Now, $t_{N-1}$ is no longer needed, so it is banished, and we have exactly one unit of free memory. To compute $\hat{t}_{N-1}$, we require $t_{N-2}$ and $\hat{t}_N$ to be in memory. Since $\hat{t}_N$ was just computed, it is clearly in memory. However, $t_{N-2}$ may or may not be in memory. We consider the two cases separately.

If $t_{N-2}$ is in memory, then we immediately compute $\hat{t}_{N-1}$. Next, tensors $t_{N-2}$ and $\hat{t}_N$ are banished, leaving us with the desired two free units of memory.

If, on the other hand, $t_{N-2}$ is not in memory, we must rematerialize it. Let $t_j$ be the resident tensor that terminates the evicted path of tensors containing $t_{N-2}$. We need to perform the sequence of computations $\{t_{j+1}, t_{j+2}, \ldots, t_{N-2}\}$. However, we only have one unit of free memory, so after computing $t_{j+1}$ we will need to evict some tensor from memory. The evicted tensor must be $t_i$ for some $i \leq j$, as neither $t_{j+1}$ nor $\hat{t}_N$ can be evicted (the former will be used for the next computation, and the latter is pinned in memory).

Regardless of which tensor $t_i$ is evicted, the length of the evicted path it creates cannot exceed $2L_S$, where $L_S$ is the length of the longest path in $S$. Lemma A.3 bounds $L_S \leq \frac{2(N-2)}{B-1}$, so this step of the algorithm maintains Item 2 of the lemma.

It remains to show that the maximum gap in $\mathcal{C}$ does not become larger than $2L_S$ during the remaining steps of rematerialization, and that the computation of $\hat{t}_{N-1}$ ends with at least two units of free memory. To show the first claim, we note that the number of evicted tensors on the path to $\hat{t}_{N-1}$ does not exceed $2L_S$ (this is the maximum length possible, if $t_j$ was evicted and its adjacent evicted paths were both of length $L_S$). Therefore, when performing the intermediate rematerializations necessary to rematerialize $t_{N-2}$, it is always possible to evict a tensor between $t_j$ and $t_{N-2}$, with a heuristic value of less than $2L_S$. Since we evict the tensor with the smallest heuristic value, we will never create an evicted path of length greater than $2L_S$.

Finally, we note that, after computing $\hat{t}_{N-1}$, both $t_{N-2}$ and $\hat{t}_N$ will be banished. This leaves us with the desired two units of free memory.

We have shown that, after computing $\hat{t}_{N-1}$, the algorithm has two units of free memory, and the checkpoint set $\mathcal{C}$ has a maximum gap of no more than $2L_S$. Next, we show that this set $\mathcal{C}$ is maintained throughout the remainder of the backward pass.

**Phase 2: The remaining $N - 2$ gradient computations.**

The analysis for the remainder of the backward pass follows via induction, using the argument for rematerializing $t_{N-2}$ above.

We have already shown a base case; we can maintain the desired properties of $\mathcal{C}$ when computing $\hat{t}_{N-2}$. For the inductive step, consider the computation of $\hat{t}_i$ for $1 < i < N - 1$. Suppose we have at least two units of free memory, and $\hat{t}_{i+1}$ in memory. Furthermore, suppose that the set $\mathcal{C}$ satisfies the properties of the lemma. We need to rematerialize $t_{i-1}$, which terminates a path of evicted tensors of length no more than $2L_S$. As we rematerialize this path, it may require evicting tensors from memory. However, by the same logic we applied above, we know that the algorithm may always choose to evict a tensor resulting in a path of less than $2L_S$. The algorithm will always choose this option in favor of creating a longer evicted path. We conclude that the upper bound of $2L_S$ is preserved when computing $\hat{t}_i$. Furthermore, after $\hat{t}_i$ is computed, we may evict $\hat{t}_{i+1}$ and $t_{i-1}$, giving us two units of free memory. This proves the inductive step.

Note that, in the case that $i = 1$, the computation requires no rematerializations, as $\hat{t}_1$ only depends on $\hat{t}_2$, and the latter is in memory at the time of computing $\hat{t}_1$. $\qquad\square$

PROOF OF LEMMA A.2

*Proof.* Let $C_{i,k}$ denote the cost of processing gradient $i$ in this group. Since there are $L_k$ associated gradients, the total cost is

$$C_k = \sum_{i=1}^{L_k} C_{i,k}.$$

To compute each $C_{i,k}$ we note that computation of the gradients proceeds in phases. When the first gradient is computed (at cost $C_{0,k} = L_k$), two units of memory must be devoted to the current tensor computation, while the remaining $k$ units of memory are used for intermediate rematerialized tensors. Applying the intermediate checkpointing lemma, A.4, we conclude that some of these intermediate tensors will remain as checkpoints (indexed by $j$, with $j = 1$ indicating the highest-indexed tensor), with adjacent checkpoints separated by a distance at most $L_{k,j} = \frac{4(L_k - 2)}{k-1}$. We can express the total cost of computing the gradients in this gap as

$$C_k = L_k + \sum_{j} \sum_{i \in \text{group } j} C_{i,k}$$

We begin by considering the first group to be processed, $j = 1$, associated with the last path between checkpoints. Since it is the first group to be processed, it has no spare memory for intermediate checkpoints. Therefore, computing the first gradient requires rematerializing the entire group (with at most $L_{k,j}$ intermediate tensors), computing the second gradient requires rematerializing at most $L_{k,j} - 1$ tensors, and so on. This gives a total cost bounded as follows (using $\lesssim$ to denote inequality up to constant factors).

$$\sum_{i \in \text{group } 1} C_{i,k} \leq \sum_{l=0}^{L_{k,j}} L_{k,j} - l$$
$$\lesssim (L_{k,j})^2$$
$$= \left( \frac{4(L_k - 2)}{k - 1} + 1 \right)^2$$
$$\lesssim \frac{L_k^2}{k^2}$$

Next, we compute the total cost of calculating all the gradients between checkpoints $j$ and $j + 1$. When the algorithm begins to compute group $j$, it has $j$ pieces of extra memory, allowing it to further subdivide group $j$ into $j + 1$ intervals. By the intermediate checkpointing lemma, each of these

intervals is of length at most $\frac{4(L_{k,j}-2)}{j-1}+1$. We have

$$
\sum_{i \in \text{group } j} C_{i,k} \leq j \sum_{l=0}^{\frac{4(L_{k,j}-2)}{j-1}+1} \frac{4(L_{k,j}-2)}{j-1} + 1 - l
$$

$$
\lesssim j \left( \frac{4(L_{k,j}-2)}{j-1} + 1 \right)^2
$$

$$
\lesssim \frac{L_{k,j}^2}{j}.
$$

Summing over the at most $k$ checkpoints $j$, we conclude

$$
C_k \lesssim L_k + \sum_{j=1} \frac{L_{k,j}^2}{j}
$$

$$
= L_k + L_{k,j}^2 H_k
$$

$$
\lesssim L_k + \frac{L_k^2}{k^2} \log k
$$

where $H_k$ is the $k^{th}$ harmonic number.    □

**Lemma A.4** (Intermediate Checkpointing). *Consider the behavior of the DTR algorithm using the heuristic $h_{e^*}$, when computing gradients for the backward pass. Suppose, immediately prior to the computation of gradient $\hat{t}_i$, we have $2 + k$ pieces of free memory ($k \geq 0$), and that $\hat{t}_{i+1}$ is in memory. Suppose also that forward tensor $t_j$ is the first resident ancestor of $\hat{t}_i$, so that we will rematerialize $t_{i-1}$ starting from $t_j$ to compute $\hat{t}_i$. Finally, suppose that $t_j$ is never evicted until it is banished.*

*Then, immediately after computing $\hat{t}_i$, memory contains a set of "checkpoint" tensors $\mathcal{C}$ with the following properties:*

1. *The tensors in $\mathcal{C}$ remain in memory until they are banished.*

2. *The gap between neighboring tensors in $\mathcal{C}$ satisfies*

$$
L \leq \frac{2((i-j)-1)}{k+1}
$$

*Proof.* We begin by analyzing the state of memory after computing $\hat{t}_i$. Since we started with $2 + k$ pieces of free memory, and rematerialized $t_{i-1}$ starting from $t_j$, Lemma A.3 tells us that, after rematerializing $t_{i-1}$, the gaps in memory between $t_j$ and $t_{i-1}$ are all bounded by

$$
L \leq \frac{2((i-j)-1)}{k+1}.
$$

We need to evict one additional item from memory, in order to compute $\hat{t}_i$. After this single eviction, the maximum gap is no more than doubled. We conclude that, after computing the first gradient, the maximum gap is no more than $2L$.

It remains to show that the maximum gap in $\mathcal{C}$ does not become larger than $2L$ during the remaining steps of rematerialization. To show this, we first note that the computation of the next gradient, $\hat{t}_{i-1}$, begins with two units of free memory (having just banished $\hat{t}_{i+1}$ and $t_i$). We also note that the number of evicted tensors that need to be rematerialized for this gradient computation does not exceed $2L$. Therefore, when performing the intermediate rematerializations necessary to rematerialize $t_{i-2}$, it is always possible to evict a tensor with a heuristic value less than $2L$. Since we evict the tensor with the smallest heuristic value, we will never create an evicted path of length greater than $2L$.

This argument can be applied for every gradient computed between $\hat{t}_i$ and $\hat{t}_{j+1}$, which shows that the desired properties of $\mathcal{C}$ are maintained.    □

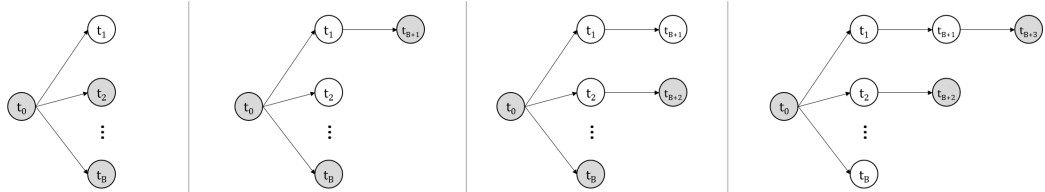

Figure 6: An example construction of an adversarial graph. Gray tensors are in memory ($t_0$ must always be in memory). The initial tensor $t_0$ has $B$ paths descending from it, so there is always some path from $t_0$ with no resident tensors. The adversarial construction chooses to place the next node at the end of such an entirely evicted path.

## B    PROOF OF THEOREM 3.2

In this section, we provide a proof of Theorem 3.2, which lower bounds the number of tensor computations required by DTR under any determinstic heuristic, compared to an optimal checkpointing algorithm.

*Proof.*  We will prove this theorem by designing an adversarially generated graph that forces DTR to repeatedly rematerialize evicted tensors. Our architecture simultaneously leverages the static planner's ability to reorder computations, to avoid repeated computation of evicted tensors.

Since DTR is a dynamic algorithm, it must choose which tensor to evict at time $T$ based only on the portion of the graph computed up to time $T$. Our adversarial architecture generator builds the network one node at a time, choosing the next node based on the previous choice of the DTR algorithm. The construction is as follows:

1. The graph begins with tensor $t_0$, which, by the behavior of DTR, must remain in memory. Tensor $t_0$ has $B$ children, $t_1$ through $t_B$.

2. After step $B$ of the computation, one of $t_0$'s children must no longer be in memory. Call this evicted child $t_*$ The next node revealed by the adversary is the child of $t_*$, causing DTR to rematerialize $t_*$.

3. The adversary continues to repeat this construction. Since $t_0$ has $B$ children, but there are only $B - 1$ units of memory to allocate among its descendants, there must be some path from $t_0$ that contains no resident tensors. The adversary reveals the next resident tensor on the end of that path, causing DTR to rematerialize the entire path. This repeats until we have revealed all $N$ nodes of the graph.

An example construction of the adversarial architecture is given in Figure 6.

Next, we analyze the computation of DTR on this graph. To do this, we sum the cost of computing each tensor $t_1$ through $t_N$. Consider the architecture of the final revealed network, and let $L_j$ denote the length of the path starting from $t_j$, where $j = \{1, \ldots, B\}$ so that $t_j$ is a direct child of $t_0$. Since our adversary places the next node such that the entire path must be rematerialized, the total cost of computing this graph dynamically is

$$C = \sum_{j=1}^{B} \sum_{i=1}^{L_j} i$$

$$= \sum_{j=1}^{B} \frac{1}{2} L_j (L_j + 1)$$

$$\approx \sum_{j=1}^{B} L_j^2$$

where $\approx$ hides constant factors. This sum is minimized when the $L_j$ are all equal, which gives $L_j = (N-1)/B$. The cost of computing all the tensors is therefore at least

$$C \gtrsim \sum_{j=1}^{B} N^2/B^2$$
$$= N^2/B$$

To finish the proof, we upper bound the cost of the optimal static algorithm on this adversarial graph by exhibiting one static checkpointing algorithm and analyzing its behavior. The static algorithm may observe the entire structure of the $N$ nodes, and rearrange the computation in any equivalent order.

Consider the static algorithm that computes the entire graph one path at a time. That is, the algorithm first computes $t_1$ and all its children (requiring only two units of memory, with no rematerializations), then computes $t_2$ and all its children (again, reusing the same two units of memory), until all $B$ paths are computed. The total cost is therefore $\Theta(N)$.

We see that DTR requires $\Omega(N^2/B)$ computations to compute the tensors in this graph, whereas a static checkpointing algorithm would only require $\Theta(N)$ computations. We conclude that when DTR is run with a deterministic heuristic, there exists an architecture on which it requires at least $\Omega(N/B)$ times the runtime of a statically checkpointed evaluation. $\qquad\square$

## C  Simulator Specification

In this section, we provide a detailed technical specification of the DTR simulator. This includes fundamental abstractions, formal definitions of heuristics, pseudocode, runtime optimizations, and details about the log-replaying mechanism.

### C.1  Fundamental Abstractions

We designed the simulator to support computations logged from PyTorch (see Sec. C.6). In PyTorch, a tensor is a view (containing metadata) of a buffer; multiple tensors can point to a single buffer. This allows us to model the various aliasing relations between tensors in PyTorch (Paszke et al., 2017); other DL frameworks likely also use a similar representation.

**Storage.**  At its core, DTR is a runtime system for reducing memory usage. As such, *storages* (*i.e.,* buffers of memory) are the underlying unit which DTR operates on. They support the following operations:

- $size$ : **Storage** $\to \mathbb{N}$: the size of the storage in bytes;
- $root$ : **Storage** $\to$ **Tensor**: the tensor whose parent operation computes the contents of the storage (there is exactly 1 for each storage);
- $tensors$ : **Storage** $\to$ **List**[**Tensor**]: all tensors which view the storage;
- $resident$ : **Storage** $\to$ **bool**: true iff the storage is in memory;
- $locks$ : **Storage** $\to \mathbb{N}$: the number of locks on the storage held interally by DTR (indicating the storage is needed for pending rematerializations);
- $refs$ : **Storage** $\to \mathbb{N}$: the number of external references to the storage, *i.e.*, those held by user code.

We say a storage $S$ is *evictable* if and only if $resident(S) \wedge locks(S) = 0$.

**Tensor.**  Each tensor $t$ has an associated "parent" operation $op(t)$ which computes it (potentially along with $storage(t)$, its underlying storage).

Each tensor $t$ also has an external reference count $refs(t)$; in particular, each storage $S$ has $refs(S) = \sum_{t \in tensors(S)} refs(t)$. The external reference count is used to track whether a tensor is still live in the source program or whether it should be treated as having been deallocated by the source

program. Additionally, $t$ is an *alias* iff $t \neq root(storage(t))$, meaning that $t$ is a view of a storage created by a different parent operator. For convenience, we define $size(t)$ to be 0 if $t$ is an alias and $size(storage(t))$ otherwise (since the metadata will likely be on CPU).

Unlike storages, a tensor $t$ is resident when $storage(t)$ is resident *and* $op(t)$ has been performed *after* $storage(t)$ last became resident. This condition is denoted as $defined(t)$, and models the behavior of our PyTorch prototype implementation where the whole tensor object is destroyed upon storage eviction (including metadata about the view, like striding and offset)[2]. Thus, before an operation depending on $t$ can be executed, $defined(t)$ must be satisfied, given our assumption that views of a storage must be evicted once the underlying storage has been evicted. Note that for a non-alias tensor $t$, we have $resident(storage(t))$ if and only if $defined(t)$.

**Operator.** An *operator* represents a fundamental unit of computation in DTR. Operators are assumed to be pure functions of their arguments, not depending on any other external state (see Sec. C.6 for our handling of mutation). As such, each operator $f$ has an associated compute cost $cost(f) \in \mathbb{N}$. We assume each $f$ has type **List[Tensor]** $\rightarrow$ **List[Tensor]** and define $inputs(f)$ and $outputs(f)$ to be the input and output tensors of $f$, respectively.

## C.2 Formal Metadata Definitions

While our abstract description of DTR in Figure 1 is over tensors, the simulator operates over storages rather than tensors. Thus we must define the metadata our heuristics use over storages, providing notions of cost, staleness, and data dependencies for storages rather than for tensors.

**Cost.** For a given storage $S$, we define the compute cost of $S$ as

$$cost(S) := \sum_{t \in tensors(S)} cost(op(t)).$$

This is a worst-case estimation: it represents the compute cost which is incurred when every tensor view of $S$ needs to be rematerialized. An alternative definition is simply $cost(op(root(S)))$, which may be acceptable as aliasing operations are typically much cheaper than non-aliasing.

**Staleness.** We estimate the staleness of $S$ by tracking the *last access* time of each $t \in tensors(S)$. The last access time $last\_access(t)$ is defined as the most recent time when $t$ was referenced by a queued operation. Naturally, we define $last\_access(S) = \max_{t \in tensors(S)} last\_access(t)$. Staleness, given the current time $\mathcal{T}$, is then defined as $stale_{\mathcal{T}}(S) := \mathcal{T} - last\_access(S)$.

**Data dependencies.** The dependencies of $S$ are the set of storages

$$deps(S) := \{storage(u) \mid \exists t.\, t \in tensors(S) \land u \in inputs(op(t))\} \setminus \{S\}.$$

Note that we exclude $S$ since it is not a true dependency (each alias tensor in $tensors(S)$ technically "depends" on $S$). Another possible approximation of the above is to simply take the dependencies of $root(S)$; although this ignores potential dependencies of aliasing operations, it is precise if all aliasing operations depend only on $S$.

We now define the *dependents* of $S$ as the set $deps^{\top}(S)$ consisting of all $T$ with $S \in deps(T)$. With this definition, DTR can operate over the dependency graph $(V, E)$ where $V$ is the set of storages and $(S, T) \in E$ iff $S \in deps(T)$. Note that $(V, E)$ is implicitly indexed by time $\mathcal{T}$, with $V$ being the set of non-banished but at-least-once computed storages at $\mathcal{T}$ and $E$ being the dependency relations at $\mathcal{T}$.

**Evicted neighborhood.** The *evicted neighborhood* $e^*$, as defined in Section 2, works without modification over the storage dependency graph. We define it here for completeness. Let $deps_e(S)$

---

[2]The storage field in a PyTorch tensor is immutable; in principle, we could have changed this to permit reassigning views of evicted storages to point to null and ensure the storages are rematerialized when needed, but this would have required much more extensive modifications to the codebase, which may rely on the invariant of immutable storage pointers.

be the evicted subset of $deps(S)$, and likewise for $deps_e^\top(S)$. Now, let $D_e$ and $D_e^\top$ be the transitive closures of the relations

$$\{(T, S) \mid T \in deps_e(S)\} \quad \text{and} \quad \{(S, T) \mid T \in deps_e^\top(S)\},$$

respectively. Then, $e^*(S) := \{T \mid (T, S) \in D_e\} \cup \{T \mid (S, T) \in D_e^\top\}$. Intuitively, $e^*(S)$ is the set of evicted storages that must be resident to compute all $t \in tensors(S)$, together with the set of evicted storages $T$ that need $S$ to be resident before all $t \in tensors(T)$ can be computed.

**Relaxed (Union-Find) evicted neighborhood.** Actually tracking $e^*(S)$ can be computationally expensive due to the directed and changing nature of the graph. For each $S$, $e^*(S)$ depends on its specific ancestors and descendants, which can vary as tensors are evicted and rematerialized. An exact solution would likely involve a dynamic graph connectivity data structure, which would greatly increase the complexity of the simulator's implementation.

We find an approximate solution by relaxing the definition of the evicted neighborhood. At a high level, our solution works as follows: given a storage dependency graph $G = (V, E)$, we first forget edge directions to obtain the undirected dependency graph $\tilde{G}$. Now, let $\tilde{G}_e$ be the subgraph obtained by removing all resident storages (and any edges including them). Each connected component of $\tilde{G}_e$ is then an *evicted component*, with each evicted $T \in V$ belonging to exactly one component $\epsilon^*(T)$.

Importantly, we track these evicted components using a *Union-Find* (UF) data structure, which efficiently supports merging and obtaining static set metadata. Each component tracks the sum of the compute costs of its elements (with the union of two components having the sum of each constituent cost). We denote the associated UF set for a storage $T$ by $T.set$, which is mutable state.

We can now define the relaxed evicted neighborhood for a resident storage $S$ as

$$\tilde{e}^*(S) := \left( \bigcup_{T \in deps_e(S)} T.set \right) \cup \left( \bigcup_{T \in deps_e^\top(S)} T.set \right).$$

Note that in practice, no UF unions are performed when querying this approximation. Instead, we collect and merge the set metadata separately, as otherwise we would erroneously merge evicted components during heuristic evaluation. This approximation reduces the worse-case time complexity of querying compute costs over the neighborhood to be linear in the number of adjacent storages, as opposed to all ancestor and descendant storages.

However, rematerializing a tensor in an evicted component creates a split in the component and *splitting* is not a supported operation on UF data structures.[3] Approaches to splitting would also need to recover the original compute costs of each set, which may require traversing the whole set if done naively. To handle splitting more efficiently, we use the following approximation: when a (previously) evicted storage $S$ is rematerialized, we first set $S.set.cost := S.set.cost - cost(S)$, and then assign $S.set := \emptyset$ (*i.e.*, assign $S$ to a new empty UF set). Note that when a storage is first computed, its evicted component is also initialized to be empty. While resident storages thus never count towards the compute cost of a component, "phantom connections" between evicted storages may accumulate over time (likely depending on the connectedness of the underlying dependency graph). Despite this limitation, this approximation worked well in practice, as seen in the simulated and prototype results.

## C.3 Formal Heuristic Definitions

Having defined the metadata above, we can now formally define the $h_{\text{DTR}}$ variants used in Sec. 4. (Recall that $h_{\text{DTR}}$ heuristics compute a score using measures of size, computational cost, and staleness and evict the tensor with the smallest score, corresponding to the intuition that the tensor evicted should be large, unlikely to be rematerialized, and cheap to rematerialize if it does need to be rematerialized.)

$$h_{\text{DTR}}(S) := \frac{cost(S) + \sum_{T \in e^*(S)} cost(T)}{size(S) \cdot stale_\mathcal{T}(S)}.$$

---

[3]This can be seen as a variant of the Union-Find-Split problem, which typically requires the use of more complex data structures such as link-cut trees.

$$h_{\text{DTR}}^{\text{eq}}(S) := \frac{cost(S) + \sum_{T \in \tilde{e}^*(S)} cost(T)}{size(S) \cdot stale_{\mathcal{T}}(S)} \approx \frac{cost(S) + cost^*(S)}{size(S) \cdot stale_{\mathcal{T}}(S)}$$

Note that the simulator implementation uses the splitting approximation described above, with $\tilde{e}^*(S)$ depending on the specific sequence of evictions and rematerializations. $cost^*(S)$ in the second expression is used to denote this statefulness.

$$h_{\text{DTR}}^{\text{local}}(S) := \frac{cost(S)}{size(S) \cdot stale_{\mathcal{T}}(S)}.$$

## C.4 IMPLEMENTATION DETAILS

**Runtime state.** In what follows, we denote the collective runtime state of the DTR simulator as $R$, and use the dot notation to indicate stateful reads and writes of runtime values. The simulator tracks the following runtime state:

- $R$.heuristic : $(\mathbf{Storage}, \mathbf{Metadata}) \to \mathbb{R}$, the eviction heuristic, interpreted as a score (the lowest-scored storage is evicted);
- $R$.budget : $\mathbb{N}$, the memory budget in bytes;
- $R$.memory : $\mathbb{N}$, the current memory usage in bytes;
- $R.\mathcal{T}$ : $\mathbb{N}$, the current clock time in some unit of granularity, such as nanoseconds;
- $R$.pool : $\mathbf{List}[\mathbf{Storage}]$, list of all currently evictable storages.

**Eviction and banishing.** To evict a given storage $S$, we set all tensors in $S$ to be undefined, remove $S$ from the pool, and decrease $R$.memory by $size(S)$. Cached metadata are also updated as necessary.

Banishing (*permanent* eviction) is slightly more subtle; in particular, it can only be done for $S$ when $deps_e^\top(S) = \emptyset$. Banishing then proceeds by evicting $S$ as above, but with the additional effect of removing $S$ entirely from the dependency graph. Each $T \in deps^\top(S)$ is then locked (and effectively becomes an non-rematerializable constant). Storages locked in this way are said to be *pinned* (and have a special flag in the simulator), to distinguish them from those locked during rematerialization, and we permit them to be banished in the future. Note that banishing can be performed on evicted $S$ when the above condition is met, in which case the eviction is skipped.

**(Re)materialization.** When a tensor $t$ is to be (re)materialized, its parents' storages are first locked by incrementing the lock count (so that they don't get evicted while they are still needed) and undefined parents are recursively rematerialized. We then increment $R$.memory by $\sum_{u \in outputs(op(t))} size(u)$ (performing evictions as necessary), and move $R.\mathcal{T}$ forward by $cost(op(t))$. Multi-output operations must be handled carefully so as to not leak memory: we make sure to *decrease* $R$.memory by $size(u')$ for each $u' \in outputs(op(t))$ that was defined *prior* to the rematerialization. This models the immediate freeing of doubly-computed ephemeral tensors in the PyTorch implementation. Lastly, locks on parent storages are freed and unlocked storages (including any newly rematerialized ones) are added back into $R$.pool.

**Constants.** The simulator models non-rematerializable constants like weights and inputs by creating dummy "constant" tensors using nullary operators with 0 cost and pinning the resulting storage. This allows the simulator to have a full picture of the computation graph. Furthermore, log-accurate banishing requires knowledge of constants (as PyTorch reference-counts constants).

## C.5 ADDITIONAL RUNTIME OPTIMIZATIONS

**Banishing and eager eviction.** When the final external reference to a storage $S$ is lost, we know that the underlying DL framework would have reclaimed the memory used by $S$. To utilize this information as opposed to doing nothing, DTR can either banish $S$ or simply evict $S$ normally. When banishing, the runtime must first check that $S$ has no evicted dependents; if it does, then we retry

banishing each time a dependent is rematerialized. Banishing has the ability to free constants, but at the downside of pinning potentially exploding amounts of memory. The alternative (*eager eviction*) is easier to implement and simply involves evicting $S$ normally (if possible). This prevents the problem of over-pinning memory, but with the downside that constants can never be evicted. In practice, eager evictions have allowed us to support lower budgets by pinning fewer values (see Sec. D.2 for details).

**Caching metadata.** To avoid costly recomputations of metadata during heuristic evaluations, we cache the local cost $cost(S)$ for each $S$, as it only changes when new aliases are made. Additionally, for the $h_{\text{DTR}}$ heuristic, we avoid recomputing $e^*(S)$ at each evaluation by caching and only recomputing it after evictions or rematerializations that directly affect $e^*(S)$. Such recomputations are further optimized by tracking the evicted ancestors and descendants separately (allowing them to be recomputed independently, depending on the position of the affected storage).

### C.6 Log-Replaying Mechanism

**Log format.** We logged PyTorch operations as a sequence of abstract instructions corresponding to the semantics of the actions we were easily able to instrument in the framework. Every PyTorch tensor is given a unique identifier string upon creation, which is recorded and used in the log. In this section, each PyTorch tensor $t$ corresponds to a simulator tensor $[\![t]\!]$.

The log contains the following instructions:

- MEMORY($t, size$): logs that $t$ uses $size$ memory; treated as 0 if $[\![t]\!]$ is an alias.
- ALIAS($t_o, t_i$): logs that $[\![t_o]\!]$ is an alias of $[\![t_i]\!]$, *i.e.*, two different views of the same storage. $t_i$ can either be a tensor identifier or $\bot$; if $t_i = \bot$, then $t_o$ does not alias another tensor ($t_o$'s parent operation created its storage).
- CALL($inputs, outputs, cost, op$): logs the operator call $outputs = op(inputs)$ with compute cost $cost$. This instruction is followed by $|outputs|$ MEMORY and ALIAS instructions to log information about each output. Each CALL corresponds to a simulator operator $[\![op]\!]$ with inputs $\{[\![i]\!] \mid i \in inputs\}$ and new simulator tensor outputs $\{[\![o]\!] \mid o \in outputs\}$.
- MUTATE($inputs, inputs', cost, op$): logs the in-place (mutating) operator call $op(inputs)$ with compute cost $cost$, which modifies $inputs' \subseteq inputs$.
- CONSTANT($t$): logs that $[\![t]\!]$ is a constant, and is followed by a MEMORY instruction.
- COPY($t_o, t_i$): logs a new identifier $t_o$ with $[\![t_o]\!] = [\![t_i]\!]$. This increments $refs([\![t_i]\!])$. This happens when Python code like "x = y" is called where y is a PyTorch tensor and x is a fresh variable; this action neither creates a new storage nor a new view but only has x *point to* the same view as y.
- COPYFROM($t_o, t_i$): logs the PyTorch code $t_o = t_i$ where each side is an existing tensor. This decrements $refs([\![t_o]\!])$, increments $refs([\![t_i]\!])$, and updates $[\![t_o]\!] \mapsto [\![t_i]\!]$. Intuitively, this corresponds to Python code like "x = y" where y is a PyTorch tensor and x was already assigned to a PyTorch tensor; in PyTorch, x is mutated to match y.
- RELEASE($t$): logs the destructor of the PyTorch tensor $t$. This decrements $refs([\![t]\!])$.

**Supporting mutation.** To support mutation from in-place operators, the simulator adds a "reference layer" that mutates cloned tensors, allowing for a uniform interface for all operators. Given a mutation instruction MUTATE($inputs, inputs', cost, op$), let $i_{new}$ be a new unique identifier for each $i \in inputs'$, and let $inputs'_{new} = \{i_{new} \mid i \in inputs'\}$. We then proceed by treating $op$ as a pure operator from $inputs$ to $inputs'_{new}$, where each newly created simulated tensor $[\![i_{new}]\!]$ is non-aliasing and has size $size(storage([\![i]\!]))$. Lastly, we decrement $refs([\![i]\!])$ and update the mapping $[\![i]\!] \mapsto [\![i_{new}]\!]$. Intuitively, we are modeling the transformation

$$op(t) \rightsquigarrow \texttt{Tensor } t' = copy(t); op(t'); t = t'.$$

Note that in our prototype implementation, a mutation of $i$ may produce incorrect results when $[\![i]\!]$ is an alias, since the mutation layer would create a clone but aliases would still point to the old storage. Potential solutions in real implementations would be to propagate the above rewrite to all aliases of a storage (costly) or to mutate storage pointers (which would have increased the complexity of our modfications to PyTorch).

**Output condition.** All live tensors at the end of a log (i.e. all $t$ with $refs(t) > 0$) are treated as necessary outputs (namely, gradients, the loss value, and the prediction). They are thus rematerialized (if evicted) and locked to ensure they persist. This prevents the simulator from incorrectly reporting better results by evicting computed weight gradients and never rematerializing them. This permits the user to perform the weight update step outside of DTR immediately after the backward pass ends. Based on our observations of PyTorch's `optimizer` gradient updates, we could also support performing these updates within DTR, since a parameter update simply performs in-place mutating additions (`add_`) of scaled gradients to the parameters.

## D  ABLATION STUDY

In this section, we present an ablation study comparing the impacts of different sources of information for the the $h_{\text{DTR}}$ heuristic. In addition to comparing the overhead in terms of additional tensor computations, we also consider the runtime overhead of different $h_{\text{DTR}}$ configurations in terms of the number of tensor accesses by heuristic computations and metadata updates. We also compare different eviction policies for the $h_{\text{DTR}}$ heuristics: ignoring deallocations, eager eviction, and banishing. These trials were performed using the same logs as in Sec. 4.

### D.1  DATA SOURCES

First, we will analyze the three sources of information (metadata) for the $h_{\text{DTR}}$ heuristic. Let us consider a parameterized version of $h_{\text{DTR}}$ defined as $h'_{\text{DTR}}(s, m, c)(t) = c(t)/[m(t) \cdot s(t)]$, where $s$ is a measure of staleness, $m$ is a measure of size, and $c$ is a measure of compute cost. For this study, we take $s$ and $m$ to be the staleness and size functions defined in Appendix C. For compute cost $c$, we compare the following alternatives (see Appendix C for definitions): the full $e^*$, the approximation $\tilde{e}^*$, and the local cost (cost of the parent operator only). We allow each measure to be entirely ablated (*e.g.*, $s(t) = 1$, which we denote $s = $ no).

In the following figures, we specifically have $s, m \in \{\text{yes}, \text{no}\}$ and $c \in \{e^*, \texttt{EqClass}, \texttt{local}, \text{no}\}$. Each figure fixes a choice of $c$, varying $s$ and $m$.

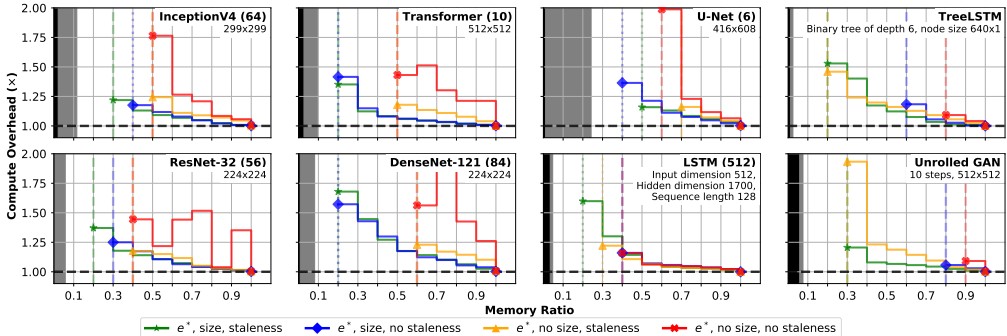

Figure 7: Results for fixed $c = e^*$, varying $s$ and $m$.

The general trend shown in Figures 7, 8, 9, 10 is that higher metadata complexity (corresponding to more precise notions of the evicted neighborhood) enables more savings, while staleness and size are required for acceptable computational overhead. It is interesting to note that the importance of staleness and size depends on the specific model architecture. For example, cost and size alone each do far better than using both cost and staleness for the static models (DenseNet, ResNet, UNet), whereas the opposite is true for the dynamic models. This may be due to model depth or the distribution of tensor sizes or to the increasing impact of individual checkpoints at lower budgets; further research may shed more light on the influence of model-specific characteristics like these. Additionally, we may note that the $\tilde{e}^*$ approximate cost performs comparably to the $e^*$ exact cost while requiring less information, validating our claim that the equivalence classes are a useful approximation.

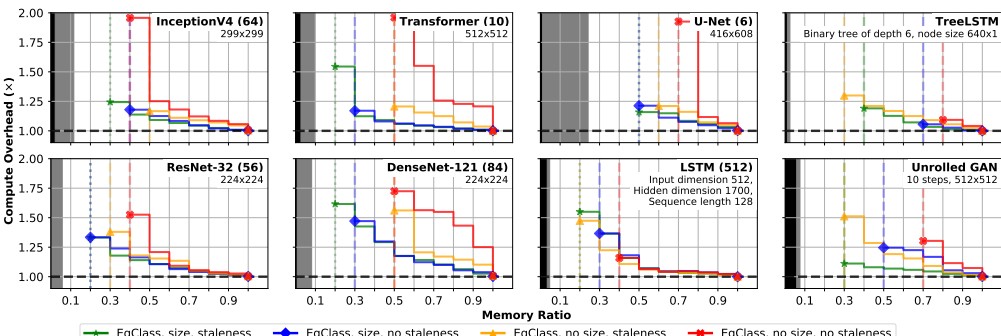

Figure 8: Results for fixed $c = \texttt{EqClass}$, varying $s$ and $m$.

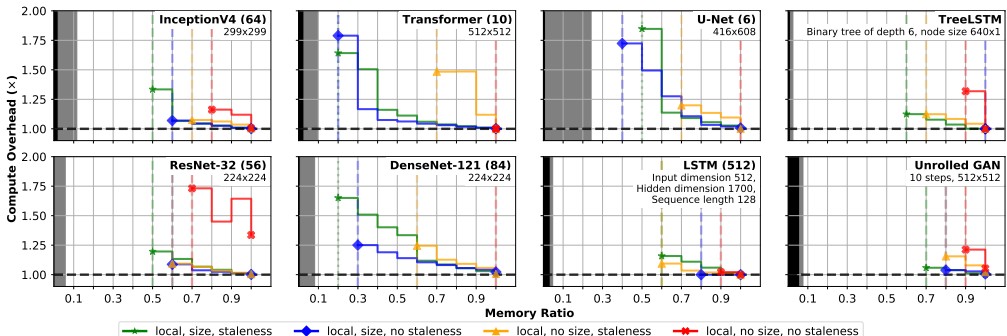

Figure 9: Results for fixed $c = \texttt{local}$, varying $s$ and $m$.

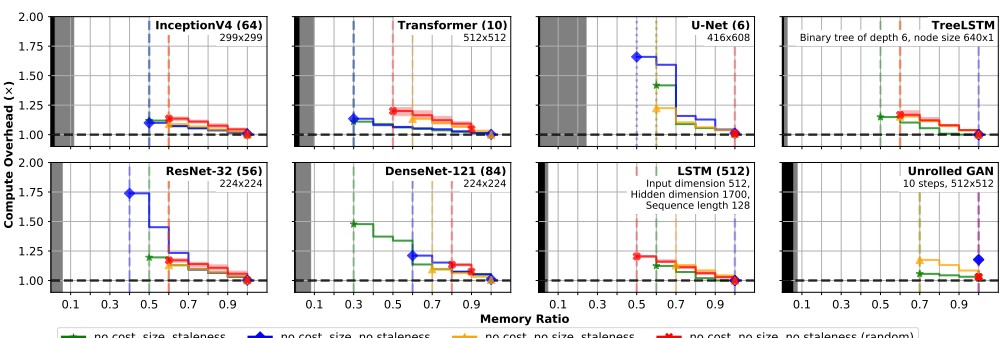

Figure 10: Results for fixed $c = \text{no}$, varying $s$ and $m$.

In general, the best-performing of these heuristics were those with non-ablated choices of $s$, $m$, and $c$, hence our choosing the $h'_{\text{DTR}}$ variants with $e^*$, $\tilde{e}^*$, and local cost ($h_{\text{DTR}}$, $h_{\text{DTR}}^{\text{eq}}$, and $h_{\text{DTR}}^{\text{local}}$, respectively) for the evaluation in Sec. 4.

## D.2  BANISHING AND DEALLOCATIONS

For the following trial, we compared the $h_{\text{DTR}}$ heuristic with banishing (permanent removal) against that with eager evictions, as described in Appendix C.5. We also compare both deallocation-aware approaches against simply ignoring deallocations. We only used $e^*$ cost because it performed much better than local cost and because it would have been more complicated to update the definition of $\tilde{e}^*$ to account for banished neighbors.

The results are shown in Figure 11.

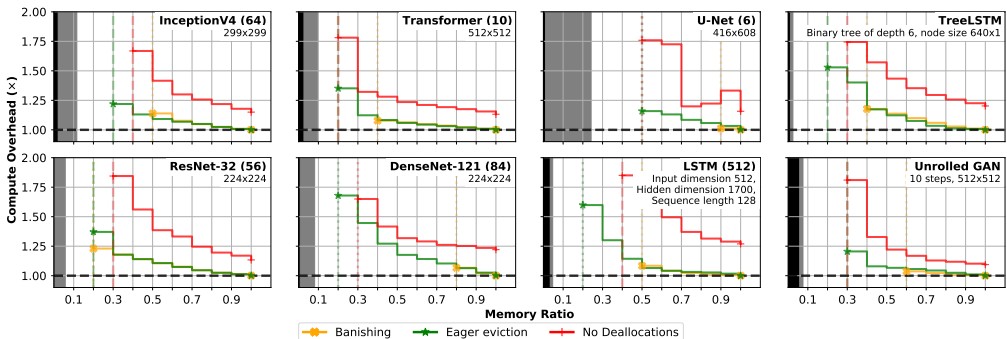

Figure 11: Results for the $h_{\mathrm{DTR}}$ heuristic, comparing banishing and eager evictions.

As the curves show, banishing is not able to achieve the same budgets across most models tested as eager eviction. For UNet, the difference is large: banishing can only support 90% of the baseline budget (and OOMs at 0.8 ratio), while eager eviction can support 50% of the baseline budget. However, banishing still attains low budgets on most models, even obtaining better computational overhead under the same budget and savings for ResNet. Since banishing potentially allows for greatly lowered runtime overhead, implementations of DTR can consider conditionally enabling it in situations where the tradeoff is more desirable.

Compared to ignoring deallocations, both banishing and eager eviction obtain noticeably lower rematerialization overhead. This shows that valuable information is captured by deallocations, and that DTR can make good use of it.

### D.3    RUNTIME OVERHEAD

For this experiment, we tracked the number of storage (see Appendix C.1) accesses made during evaluations of heuristics and maintenance of metadata. We chose this metric over wall-clock time, since our Python implementation of the simulator is not heavily optimized and may not accurately correspond to the real performance of the runtime. Storage accesses, on the other hand, do reflect operations that would be performed by a real implementation. For the $h_{\mathrm{DTR}}$ heuristic, this included each storage visited during the updating and rebuilding procedures for maintaining $e^*$ for resident storages. For the $h_{\mathrm{DTR}}^{\mathrm{eq}}$ heuristic, this included each storage visited whenever the Union-Find data structure was traversed for each evicted component (which occurs mainly during merging and when reading the compute cost). The $h_{\mathrm{DTR}}^{\mathrm{local}}$ heuristic does not need to maintain any non-local metadata. For all heuristics, each heuristic evaluation counted as one storage access.

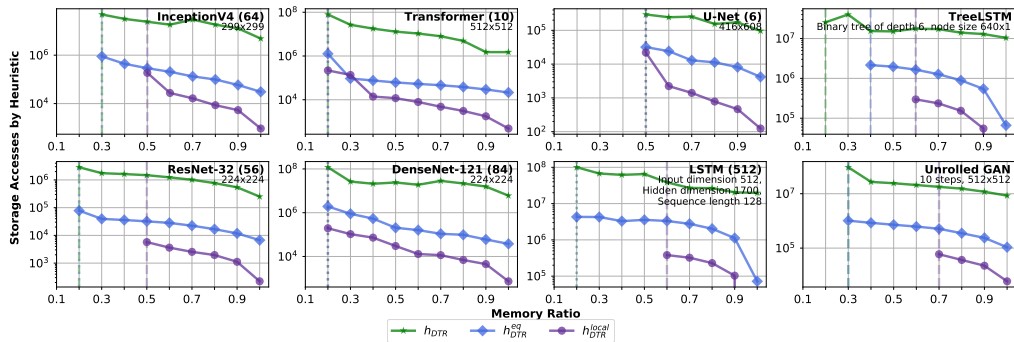

Figure 12: Total storages accesses incurred by heuristic evaluations and metadata maintenance, compared across different memory ratios, for the 3 main $h'_{\mathrm{DTR}}$ variants.

As Figure 12 shows, the accesses made by each heuristic are generally separated by at least an order of magnitude. This confirms our intuitions about the runtime overhead of each heuristic, and supports

our choice of $h_{\mathrm{DTR}}^{\mathrm{eq}}$ as a good middle ground (in terms of both runtime and computational overhead). However, these overhead figures could be improved with better-optimized implementations of the heuristics, as our implementation recomputes heuristics often, even when it may be possible to store the scores for tensors and maintain them in a sorted order. (Reformulating staleness to avoid having to use the current time might help.) Using persistent data structures that can be incrementally updated and maintain a sorted order will make these heuristics much more efficient, though this would also increase the complexity of the implementation.

## E    PROTOTYPE IMPLEMENTATION

### E.1    INTEGRATION INTO PYTORCH

To avoid modifying PyTorch's core systems, our DTR prototype is implemented as a wrapper over PyTorch's existing tensor implementations. Namely, we add a new tensor representation into PyTorch called a `CheckpointTensor`, which is simply a wrapper over an existing PyTorch tensor that additionally tracks the tensor's parent operation and other metadata (such as the last access time and the cost of the parent operation, which is timed when the tensor is first created) and registers the tensor in the DTR runtime system. Timing operators for metadata purposes simply uses the system clock, hence to guarantee the correctness of these operator times, we force PyTorch into synchronous execution mode (which ensures that GPU operators are performed synchronously); we found that DTR was still able to execute models on greatly reduced memory budgets without turning on synchronous execution mode, even though this should skew DTR's recorded operator times.

For evictions, `CheckpointTensors` are capable of freeing their underlying tensor representation from memory; they keep a closure for replaying the parent operation, which the runtime can invoke when the tensor must be rematerialized. To handle deallocations by the original program, `CheckpointTensors` also report increments and decrements to the reference count *of the underlying tensor* to the DTR runtime. We add a method to tensors called "`checkpoint()`" that lifts any tensor into a `CheckpointTensor` and a method "`decheckpoint()`" that extracts the underlying tensor from a `CheckpointTensor`, rematerializing it if necessary (we use the latter in our trials to ensure the loss and output are in memory at the end).

Our modified version of PyTorch dispatches any operation involving a `CheckpointTensor` to a specific implementation for `CheckpointTensors`; this is the same mechanism that Py-Torch uses, for example, to dispatch operations on GPU-managed tensors to CUDA implementations. Specifically, whenever PyTorch encounters an operator where an argument is a `CheckpointTensor`, its dispatch mechanism searches for a specific overload of that operator for `CheckpointTensors`. Since a `CheckpointTensor` simply wraps the underlying Py-Torch tensor, adding `CheckpointTensor` implementations for operators simply requires invoking the operator's existing implementation for the underlying tensor and wrapping the result in a `CheckpointTensor`. These overloads were essentially boilerplate code and it is likely possible to generate them automatically. As far as PyTorch's dispatch system is concerned, all tensor accesses occur through operators, so updating metadata like access time only reqires invoking the DTR runtime inside the `CheckpointTensor` operator overloads.

The DTR runtime is simply a singleton that keeps a pool of all `CheckpointTensors` created since the start of the program. The runtime is also responsible for maintaining the equivalence class data structure needed for $h_{\mathrm{DTR}}^{\mathrm{eq}}$, described in Appendix C.1 (updated each time a `CheckpointTensor` is evicted or rematerialized). Before each `CheckpointTensor` operation, the DTR runtime checks whether the memory budget has been exceeded; if it has, the runtime searches over the pool of `CheckpointTensors`, computing the heuristic score ($h_{\mathrm{DTR}}^{\mathrm{eq}}$) for each using their metadata, and evicting the least-scoring until either it is not possible to evict any more tensors or the budget has been met. (*N.b.*, this means that the prototype permits exceeding the budget by exactly one tensor allocation. In principle, we can correct this by inserting a callback into PyTorch's GPU memory manager to call the DTR runtime as soon as an allocation is *requested*; we did not do this to simplify our implementation.) This method of searching is very simplistic; it is likely that redundant heuristic computations can be removed using data structures to keep `CheckpointTensors` in a sorted order and incrementally update metadata, but the optimizations discussed below in Appendix E.2 were very simple and helped to reduce some of the overhead from this naive method. The DTR runtime is also responsible for implementing the logging mechanism described in Appendix C.6; this is

accomplished by simply writing JSON records of events intercepted by the runtime (operator calls, reference count increments and decrements, *etc.*) to a file.

The DTR prototype supports PyTorch's implementation details like in-place mutations, aliasing, and multiple operator outputs, which are all discussed in Paszke et al. (2017), using the same methods as the DTR simulator (see Appendix C). As in Appendix C.6, the DTR prototype supports PyTorch operators that perform in-place mutations by introducing a copy-on-write mutation layer: The mutating operator is made pure (and therefore infinitely replayable) by copying the source tensor for the mutation and mutating the copy. (Similarly, impure operators like `batchnorm` and `dropout` are made pure by treating state like the PRNG seed as part of the input to the operators and the updated state as part of their output.) The DTR runtime performs these copies for `CheckpointTensor` operator overloads to mutating operators. To support operators whose results are aliases of their arguments, the DTR runtime groups together all `CheckpointTensor`s whose underlying tensors are aliases of each other into *alias pools*. When a member of an alias pool is evicted, all members of the alias pool are treated as evicted; aliases are, however, rematerialized separately, only as they are needed. For `CheckpointTensor`s produced by multi-output operations, the DTR runtime allows them to be evicted separately but ensures that they are rematerialized together.

## E.2    RUNTIME OPTIMIZATIONS

Searching for tensors to evict is a significant source of overhead for DTR's runtime because the runtime recomputes each tensor's staleness and equivalence class cost upon each eviction, rather than storing and incrementally updating this information. In principle, we could reduce this portion of the overhead by using more complex data structures to maintain an ordering of the tensors to avoid searching, though this would greatly increase the complexity of our implementation. As a simpler means of reducing the DTR runtime's overhead from searching and computing heuristic scores, we added two approximate optimizations to reduce the search space: ignoring small tensors (less than 1% of the average size) and only searching over a random sample of $\sqrt{n}$ tensors from the pool of $n$ evictable tensors. This greatly reduces the number of tensors that the runtime needs to check upon evicting. Even though this improves the search overhead considerably, searching and computing costs still present considerable DTR-specific overhead, as the profiling breakdown in Figure 4 shows. Additionally, random sampling caused occasional failures at low budgets or very large inputs due to excluding good eviction candidates from the search space, which led us to deactivate that optimization in certain trials. (At low budgets, individual eviction choices are very impactful, so removing tensors from the search space completely at random can dramatically affect the results.)

There are also several possible sources of runtime overhead that could potentially be improved by making deeper modifications to PyTorch's core systems. For example, we introduced an overload layer that results in many more layers of callbacks. The mutation layer also clones tensors (even though it frees the necessary space immediately), resulting in additional overhead. Further modifications to the framework could allow for more optimizations, particularly by reducing the number of heap allocations and conversions between tuples and lists. PyTorch's define-by-run nature and shallow embedding into Python also meant that much of DTR's metadata, such as the parent operator of a tensor, needed to be computed at run time (such as by creating a closure). In other frameworks that feature a compilation step, such as Glow (Rotem et al., 2018), it may be possible to eliminate much of this overhead by generating these structures in a compiler pass. We may also note that all the bookkeeping for DTR takes place on CPU while operators are generally offloaded to other devices, so an implementation could interleave these updates with GPU operations.

## E.3    HANDLING ERRORS IN TRIALS

As discussed in Table 1 and Figure 4, the DTR prototype encountered errors on certain models when running on low budgets or on large input sizes. These errors were primarily CUDA out-of-memory errors (OOMs), but in some cases, the trial simply hung, neither crashing nor terminating. For CUDA OOMs, disabling the random sampling optimization described in Appendix E.2 eliminated the errors in most cases, suggesting that the OOMs were due to excluding useful eviction candidates. For the hanging trials, we were not able to determine whether the root cause was DTR thrashing (being trapped in a very deep recursive rematerialization, as occurred in some of the simulated trials on certain heuristics) or an infinite loop or deadlock elsewhere in PyTorch; we can investigate the cause

by further instrumenting the implementation, but we have been unable to consistently reproduce hanging trials and they seem to occur less frequently than OOMs.

In the largest two batch sizes for UNet in Table 1, disabling sampling did not eliminate all OOMs or hanging trials. Thus, for the large-input trials in Table 1, we employed a procedure for retrying upon encountering an OOM or a hang. First (as with all other GPU measurements), we perform some untimed "warm-up" trials to allow for CUDA initialization and caches to be populated and then begin timing the trials. If a trial raises a CUDA OOM or hangs (which we define as taking twice as long as the trial before it), we keep the measured times from that point in the trial and then restart (doing another warm-up), collecting the remaining number of measurements. Restarting the measurement run was the only way to ensure that all memory allocated during the trial would be collected in the event of an OOM (attempts to proceed simply by resetting the PyTorch allocator's cache resulted in memory accumulating between trials regardless). Our experimental setup automates this process of retrying failed trials and reports the total number of retries. Note that we treat failures during warm-up runs the same as failures in timed runs, since recovering from an OOM would require exiting the process running PyTorch and reinitializing CUDA. In the Table 1 results, there was 1 failed run for UNet on batch size 9 and 10 failures on batch size 10; most of the latter were during warm-up runs.

A possible reason for the occasional failed trails in UNet may be variance in operator timings, which affect the metadata and may be influencing rematerialization decisions. One way to control for this possibility in a static model like UNet would be to use a DTR simulation to produce a static rematerialization schedule and therefore have a known, safe execution schedule for operators. For a dynamic model, a static plan is not an option, but variations in operator timings could be reduced by using a fixed cost model for operators instead of timing them dynamically. That is, the DTR heuristics employed could be defined to use proxy measures that are less subject to variation (*e.g.*, defining staleness in terms of a counter incremented by operations rather than wall-clock time) or less likely to be influenced by specific system implementation details in order to have more predictable and reproducible behavior.

