# OpenReview forum: "Dynamic Tensor Rematerialization"
_ICLR.cc/2021/Conference — ICLR 2021 Spotlight_

### Official Review · AnonReviewer3 · 2020-10-19
**The paper proposes an implementation of tensor rematerialization for PyTorch, in order to reduce GPU memory requirements by up to 50-80% depending on model. The implementation is dynamic (does not require pre-runtime analysis) but requires between 0-75% overhead during runtime. Several heuristics are proposed and analyzed to minimize overheads while maximizing memory saved.**

**Rating:** 7
**Confidence:** 4

**Review:**

Claims:

-Better eviction heuristics provide noticeable improvement over earlier dynamic rematerialization schemes - up to 50-80% memory savings at the cost of up to 75% increase in computation

-Seamless implementation with PyTorch; user does not need to change their code - highly impactful if incorporated into PyTorch

-Upper bound analysis shows only O(N) operations are needed (constant factor more than without rematerialization)

-Several eviction heuristics are studied, including from previous literature. Mathematical formalism unifies these heuristics and clearly describes their relationships

Pros:

-It seems that previous rematerialization schemes provide up to 50% memory savings (i.e. can run 2x larger models), but the proposed work goes up to 80% memory savings (i.e. can run 5x larger models). If this is true, then the ability to run 5x larger models on the same hardware shifts rematerialization from a relatively minor optimization to a game-changing feature

-Running time overhead below 100%, which is an acceptable tradeoff (authors consider runtime overhead >100% to be thrashing, which is sensible)

-More so than typical papers, the design motivations are factually and intuitively explained, and performance claims are not exaggerated but rather put into perspective. The writing strikes me as convincing.

-Comprehensive experiment design with 8 models and 7 eviction heuristics, covering RNNs, CNNs, Residual networks, Transformers, and unconventional models such as Unrolled GAN. This leaves little doubt as to the proposed method's effectiveness

Cons:

-The paper did not study rematerialization in at least the multi-GPU, if not distributed-parallel setting. I think the latter can be excused given the typical scope of an ICLR paper, but I was expecting the former and was surprised to find the experiment design only involved one GPU. At minimum, a multi-GPU experiment is needed to confirm the implementation works with multi-GPU setups, if not distributed-parallel setups.

Questions:

-The paper does not talk about datasets used. This is appropriate if all models in the experiment suite do not have dynamic structure, since dynamic structures could (depending on the implementation) cause the computational graph to differ with different datasets - e.g. I imagine this could happen with NLP and BiLSTM or recursive neural networks. Can the authors clarify if they limited their models to those with static (fixed computational graph) structures? How would the proposed method behave on models with dynamic structures that change with the input data?

Suggestions for improvement:

-The authors call out the relationship between rematerialization and tensor swapping only in the related work. While this is better than not bringing it up at all, it can be argued that since rematerialization and swapping are both dynamic memory management techniques, swapping should at least be mentioned in the introduction. Even if swapping is out of scope of the paper, I think some discussion on how DTR and swapping could be combined would make the paper even stronger.

-Although the paper does explain and provide experiments as to why dynamic rematerialization is superior to static rematerialization, I think the argument could be made much more convincing if the authors showed evidence that static rematerialization does not work on certain dynamic models, limiting its applicability. For example, a statement to the effect of e.g. "Checkmate doesn't work on model X, and here is why..." could be instructive.

---

> ### Author Response · Authors · 2020-11-14
> **Response to review**
>
> We thank you for your careful reading and your detailed review, particularly for your questions and suggestions.
>
> ### Multi-GPU Setting
>
> Like recent checkpointing work, such as Jain _et al._ (2020), Kumar _et al._ (2019), and [Beaumont _et al._ (2019)](https://hal.inria.fr/hal-02352969/document), we have focused on a single-GPU setting. However, nothing in DTR’s design precludes supporting multiple GPUs. One way to add support would be to have the runtime system manage each GPU individually, treating values sent to a particular GPU as "inputs" (unevictable) and rematerializing values specific to that GPU to avoid incurring cross-GPU communication expenses.
>
> ### Question on Datasets
>
> We agree that the input to a dynamic model is very important because it can affect the model's control flow. To ensure controlled experiments in our evaluation of dynamic models, we were careful to specify and fix the input shape (sequences of a given length for LSTM and balanced binary trees of a specific depth for TreeLSTM) to ensure that we would not be introducing more variables into the experiment. That is, we used synthetic inputs of fixed shapes to ensure that in each batch, the models would follow the same control flow -- this was to ensure fair and clear comparisons. By design (and in practice using our prototype) DTR works regardless of the input shape and the resulting control flow, whereas static methods would need to perform planning for each distinct control flow pattern encountered.
>
> ### Comparing against Swapping Techniques
>
> We thank you for your suggestion about adding further discussion about how DTR specifically and checkpointing broadly differ from swapping techniques. Past swapping systems like Capuchin assume a static computation graph and a fixed access pattern (please also see our response to AnonReviewer5), which would prevent them from working with dynamic models. Additionally, as we note in our related work discussion, these systems first perform a profiling pass to gather information and then use the information they gather to plan a swapping schedule in advance so that they can overlap swapping with computation (crucial for obtaining good performance). It is thus not immediately apparent to us whether swapping can be handled entirely online on arbitrarily dynamic models.
>
> One way to combine DTR and swapping, given a fixed swapping schedule, would be to use DTR to replace the rematerialization schemes used by systems like Capuchin (perhaps given a constraint like treating values that will be swapped out as unevictable). It may also be possible to incorporate such swapping into DTR by treating swapping as a different kind of "eviction": instead of replaying computations, swapped-out values would be rematerialized using communication and the communication time would be treated as the "cost." Swapping would present interesting tradeoffs with rematerializations, as it would likely scale better than certain tensor operators (see our discussion of UNet’s performance with AnonReviewer4). Swapping has some other additional constraints, such as requiring that there actually be a location to swap to (which may not be the case for edge devices) and that there be enough memory for swapping there. This would be an interesting direction for future work, though central to the issue would be ensuring that the communication could be efficiently overlapped with computation. We will expand our discussion of these points in the paper.
>
> ### Comparing against Static Methods
>
> We also thank you for your suggestion about making more explicit why past techniques would not support dynamic models in general and we agree that this would be useful for readers. The fundamental reason, as you note in your question about datasets, is that the computation graph can vary between inputs on many dynamic models (and in some more experimental models, can vary based on input _values_), while static techniques require there to be a single computation graph.

---

### Official Review · AnonReviewer2 · 2020-10-26
**This submission is based on the observation that intermediate activations can be replayed (through local forward prop calculations), rather than kept in memory for backprop to reach them after the full forward path calculation. As a result, it helps memory capacity bound cases (such as large batch) by reducing the working set of DL training.**

**Rating:** 7
**Confidence:** 5

**Review:**

Contributions: a) analyzes multiple heuristics for which tensors to evict where compute overhead of rematerialization is minimal overall, b) suggested approach is just-in-time, and thus does not require any static analysis of the network. That is, unlike prior work in this area, it covers any network type with no prior knowledge, c) offers a good formal analysis of proposed heuristic in terms of its components: staleness, memory capacity and recursive replay cost - their formalization covers previously published heuristics as well.  Experimental framework is sound. And some encouraging results are shown delivering memory capacity saving of 30% to 90% with training slowdown of 2x or less.

Prior such work all required static analysis and planning of the network - and hence were of limited use. Significant contribution of this work is summed up at the end of Sec 4.3. It achieves 'as good' results as prior state-of-the-art (Checkmate, published at MLSys) - but without any prior knowledge of the model. This significantly increases the practical significance of this work. NeuIPS 2019 work of Kumar is the other often-cited work, also based on static planning, where the authors had already noted the primary limitation keeping it from getting adopted: "algorithm yields asymptotically better schedules, the schedule length and memory depend exponentially on the path width".

I am also happy to see authors offer hardware detail of their experimental platform (Figs 2 and 4).  And PyTorch software prototype should make it easier to follow through in other frameworks. There is a decent variety in the chosen set of benchmarks as well.

---

> ### Author Response · Authors · 2020-11-14
> **Response to review**
>
> We thank you for carefully reading our work and we greatly appreciate your encouraging remarks. We hope that by making the prototype and our experimental infrastructure available, we can assist future work in this area. DTR is open source and publicly available; we also hope to upstream DTR to mainline PyTorch in coming months.

---

### Official Review · AnonReviewer4 · 2020-10-29
**A useful and practical solution for dynamic checkpointing but system evaluation needs to be well strengthened**

**Rating:** 6
**Confidence:** 4

**Review:**

The paper presents an online algorithm for dynamic tensor rematerialization.  Theoretically, it shows the same asymptotic order on the memory budget and tensor operations as of the optimal static approach.  By simulation, it shows the performance matches optimal static checkpointing in a few models.  A PyTorch prototype is implemented, which shows benefits of reducing memory footprint and increased batch size comparing with basic PyTorch models without checkpointing.

Merits of the paper:
- Address an important practical problem on how and when to perform checkpointing during DL training.
- Cover a pretty comprehensive study across theory, simulation and system implementation.
- The suggested system implementation looks simple and clean.
- Clearly written paper, which is easy to understand and follow.

Places to improve:
- The need of having dynamic approach in this area is not very well motivated.  Since the computation in most of DL models is repetitive over iterations, static approach would work pretty well.  Furthermore, since many models take long to train, spending minutes on analyzing static graph and obtaining an optimal solution seems to be time well spent, comparing with a suboptimal dynamic solution.  The motivation of developing dynamic approach needs to be strengthened.

- Although some comparison with related work is done by simulation, really system evaluation is relatively weak - it only compares with the strawman PyTorch models without checkpointing.   It would be a lot more convincing with the results comparing with basic checkpointing approach (e.g., layer wise checkpointing) and some related work.

- For some of the reported models, like Unet, the approach reduces memory footprint and helps increase batch size, which however reduces  throughput.  These are probably not good examples to show the benefits of the approach.  But it might be beneficial to point out the underlying reasons for decreased throughput so readers know when the approach would/wouldn't work well and why.

---

> ### Author Response · Authors · 2020-11-14
> **Response to review**
>
> We thank you for your attentive reading of our paper and your insightful comments.
>
> ### Motivating the Dynamic Approach
>
> We believe DTR’s simplicity, flexibility, and deployability make it an attractive target for practical use, especially for rapid prototyping and experimentation. We agree that, as you describe, in the particular case of static models that take very long to train on large datasets, investing minutes or hours to generate a checkpointing plan is a reasonable tradeoff. However, as we note in our reply to AnonReviewer5, optimal planning with tools like Checkmate can take much longer to generate plans for large models (e.g., more than a day for DenseNet161). Furthermore, many interesting dynamic models are not supported by static checkpointing approaches.
>
> Additionally, as in our response to AnonReviewer5, we stress the conceptual contribution of our online algorithm: DTR is able to produce high-quality checkpointing schemes despite lacking any advance knowledge of the model (a very reduced setting). We hope that the insights of DTR will also be useful in future work; e.g., incorporating more facts into a planning heuristic could lead to even better results. (For example, a variant of DTR that learns from past batches might be able to make better eviction decisions at lower overhead than the purely online version.) Furthermore, as an approach that makes few assumptions and is capable of supporting arbitrarily dynamic models, DTR enables exploration of more diverse model architectures and learning techniques in memory-constrained settings, including training on edge devices or tasks with higher-order derivatives like metalearning. Providing more support for such applications expands the possibilities for deep learning models and applications.
>
> ### Further Comparisons against Past Work
>
> Unfortunately, we could not find a system implemented in PyTorch that could automatically support checkpointing across the variety of models we included in our evaluation of the DTR prototype. We could potentially apply PyTorch's checkpointing API to manually checkpoint some of the models in our evaluation, but this would entail significant additional engineering effort and the quality of any such manual checkpointing schemes would depend heavily on specific knowledge of the models, threatening the fairness of the comparison.
>
> In part of our evaluation, we relied on simulation as a baseline because some past systems are simply not available (see our discussion about Capuchin in response to AnonReviewer5) and others have been implemented in different frameworks, making it difficult to set up fair comparisons. By contrast, we were able to present a fair comparison against Checkmate by directly using their MLSys 2020 artifact.
>
> ### Performance on UNet (decreased throughput)
>
> Independently of any checkpointing scheme, increasing the batch size can only improve a model’s throughput so long as the tensor operator performance scales at a lower rate than the batch size (for example, by exploiting hardware parallelism). If the tensor operators scale at the same rate or a higher rate, then increasing the batch size will at most maintain the throughput and any additional computations (e.g., rematerializations) will decrease the throughput. We further analyzed the logs DTR produced from the trials in Table 1 and observed such scaling in UNet: the `cudnn_convolution_backward` operator scaled linearly over the batch sizes we tested and accounted for 49% of UNet’s total computation time. By contrast, the same `cudnn_convolution_backward` operator exhibited sublinear scaling in ResNet-1202, resulting in increased throughput (perhaps partly due to the difference in image sizes: 224x224 for ResNet and 416x608 for UNet).
>
> While this decrease in throughput is more appropriately attributed to the operators, and not to DTR, we note that it may be possible to apply systems like Halide, TVM, or Tiramisu to generate and tune operator implementations specialized to larger batch sizes and improve the throughput.
>
> We increased the batch size in Table 1 to highlight that DTR can allow for processing larger inputs in the same amount of memory. This can represent handling larger problem sizes with models or simply running larger models, though perhaps there is a better way we can demonstrate DTR's capabilities in this regard.

---

### Official Review · AnonReviewer5 · 2020-11-06
**Interesting work**

**Rating:** 6
**Confidence:** 3

**Review:**

Summary: This paper proposed a simple yet effective greedy algorithm with a new heuristics on checkpointing deep learning models so that people could train large model with restricted GPU memory budgets. The proposed method operates in an online setting and do not need static analysis of computation graph, thus could be used for both static and dynamic models. In a restricted model setting of linear forward network and equal space and time cost for each node, the author proves the proposed method could reach the same bound on tensor operation and memory budget with previous static checkpointing methods. The author also establish a theorem on tensor operation numbers between the proposed dynamical method and an optimal static checkpointing algorithm. In experiment, the author compared the proposed method with static techniques including the optimal Checkmate tool of Jain et al. (2020), showing the proposed method gives competitive performance without static model analysis in prior. The author also compared the proposed heuristics with prior arts on several static and dynamic models. Finally, the author described a prototype of PyTorch implementation of the proposed method.

Pros:
1. While goes under a limited setting, the theoretic analysis on the tensor operation and memory budget bound of the proposed method, as well as on the relationship between the proposed method and optimal static analysis method is novel and interesting. The experiment also shows the competitiveness of the proposed method by comparing to static methods.
2. The author does a great job explaining the idea, concepts, procedures and experiments.

Cons:
1. The author provides the comparison between the proposed heuristics and others with the same greedy algorithm, but it seems not to have the full comparison to other dynamic checkpointing approach(e.g. Peng et al. (2020) ). Although experiments in the paper shows competitive results with static model, the main use cases of the proposed method might still be in dynamic models as in normal static model use cases, the time overhead of static analysis could be ignored compared to actual model training time.
2. As the proposed heuristics bears some similarity with the one used in Peng et al. (2020), it would be more convincing to also have an ablation study of replacing the heuristics used in Peng et al. (2020) with the proposed one.

References:

[1] Jain, Paras, et al. "Checkmate: Breaking the memory wall with optimal tensor rematerialization." Proceedings of Machine Learning and Systems 2 (2020): 497-511.

[2] Peng, Xuan, et al. "Capuchin: Tensor-based GPU Memory Management for Deep Learning." Proceedings of the Twenty-Fifth International Conference on Architectural Support for Programming Languages and Operating Systems. 2020.

---

> ### Author Response · Authors · 2020-11-14
> **Response to review**
>
> We thank you for your detailed review and careful reading of our paper.
>
> ### Comparisons against Capuchin
>
> Unfortunately, the Capuchin implementation has not been released. Earlier we wrote to the corresponding Capuchin author, who replied that he and his coauthors hope to make the code available but have not yet done so. Note that Capuchin was implemented in TensorFlow, which would make it difficult to set up fair head-to-head performance comparisons with DTR (that is, it would be difficult to isolate the impact of the systems in question from the differences between the underlying frameworks). We also note that Capuchin’s checkpointing approach does not support dynamic models in general, as their runtime system first performs a profiling pass on the models and assumes that the computation graph and access patterns will not change between inputs. (Accordingly, the Capuchin author Dr. Shi confirmed in our correspondence that he would not expect Capuchin to support a dynamic model because it would not be able to "identify a regular tensor access pattern to make memory management policy.") DTR makes no such assumption, as we discuss in the paper.
>
> ### Utility of Applying DTR to Static Models
>
> As noted in your review, DTR has the advantage that it checkpoints both dynamic and static models but the disadvantage that it incurs runtime overhead. We contend that, in practice, DTR’s high-quality plans, produced in real time, make it a convenient choice for many deep learning scenarios. In particular, returning to Figure 3, DTR’s plans on static models are competitive with Checkmate’s optimal plans and outperform some past static approaches. During rapid prototyping and experimentation, even a few minutes or hours of static planning may be a significant drawback. Even after the model development phase, when we might be encouraged to spend hours or days finding an optimal schedule, it is unclear that optimal methods like Checkmate can feasibly checkpoint large models. For example, Jain _et al._ (2020) note that "[f]or DenseNet161 (Huang _et al._, 2017), no feasible solution was found within one day." Finally, we note that DTR can be used as a static checkpointing technique on static models by running the DTR simulator to create a checkpointing scheme. This usage of DTR would incur no runtime overhead.
>
> We emphasize the conceptual point made by our presentation of DTR: Namely, that an online algorithm with no prior knowledge of the model is capable of producing very good checkpointing schemes in real time. While the online algorithm, implemented precisely as described in the paper, is immediately useful for both static and dynamic models (and can enable exploration of more diverse model architectures), it is our hope that the insights of DTR will also be useful in broader scenarios, e.g., to inspire further improvements in static checkpointing.

---

> > ### Comment · AnonReviewer5 · 2020-11-24
> > **Thanks to the authors for the clarification**
> >
> > Based on the fact that Capuchin's code isn't available yet I agree that a full comparison to Capuchin isn't needed. My another concern on ablation study of replacing the heuristics used in Peng et al. (2020) with the proposed one still holds.

---

> > > ### Author Response · Authors · 2020-11-24
> > > **Request for clarification on ablation study comment**
> > >
> > > Thank you for your reply. Could you please clarify what additional ablation study you feel would strengthen the submission? We have included Capuchin's MSPS heuristic in our simulated evaluation in Section 4; it would be feasible to extend the ablation study in Appendix D to include variants of MSPS. We could also extend the prototype to use MSPS, though the results in Section 4 and Appendix D were what led us to use $h_{\text{DTR}}^{\text{eq}}$ in the prototype.

---

### Decision · Program_Chairs · 2021-01-07
**Final Decision**

**Decision:**

Accept (Spotlight)

**Comment:**

The paper presents an online algorithm for dynamic tensor rematerialization.  The theoretic analysis on the tensor operation and memory budget bound of the proposed method, as well as on the relationship between the proposed method and optimal static analysis method is novel and interesting.  It covers a pretty comprehensive study across theory, simulation and system implementation. In addition, the paper is well written.